# IMPROVED DENOISING DIFFUSION PROBABILISTIC MODELS WITH EFFICIENT NON-DIAGONAL COVARIANCE MODELING

## ABSTRACT

The sampling process of Denoising Diffusion Probabilistic Models (DDPMs) can be accelerated by leveraging second-order information in the form of approximations to the denoising posterior covariance. Previous attempts at using such information have used drastic (e.g. diagonal) simplifications of the covariance. These do not do justice to the peculiar statistical structure of natural images, which exhibit strong non-diagonal correlations between pixels and color channels, and a slow-decaying power-law frequency spectrum. Here, we develop a novel covariance model that captures these features. Our Kronecker-DCT (K-DCT) model uses a Kronecker-factored decomposition of inter-color covariances and spatial covariances modeled in the frequency domain using the Discrete Cosine Transform (DCT). The use of the DCT reduces the computational complexity from quadratic to log-linear, resulting in negligible computational and memory overhead in the sampling process. By learning K-DCT-structured amortizations of the denoising posterior covariance using pre-trained score models on CIFAR-10, Celeb-A, and ImageNet datasets, we show improved performance both in terms of FID and likelihoods compared to previous SOTA denoising samplers.

## 1 INTRODUCTION

Denoising Diffusion Probabilistic Models (DDPMs; Ho et al., 2020; Song et al., 2021; Turner et al., 2024) are a family of generative models used ubiquitously for image generation (Rombach et al., 2022; Esser et al., 2024; Betker et al., 2023), where they give state-of-the-art performance both in terms of fidelity (quality of samples) and mode-coverage (sample diversity). These models sample new images by running a so-called 'denoising' Markov chain, starting from pure noise. Given the current image $\boldsymbol{x}_t$ at time step $t$, a slightly less noisy image $\boldsymbol{x}_{t-\delta}$ is obtained by sampling from a Gaussian approximation to the 'denoising posterior' $p(\boldsymbol{x}_{t-\delta}|\boldsymbol{x}_t)$, under a certain probabilistic model that defines their joint distribution. Most research efforts so far have focused on approximating the first moment of this posterior, i.e. the conditional mean $\mathbb{E}[\boldsymbol{x}_{t-\delta}|\boldsymbol{x}_t]$, using deep networks trained through various objectives. The main justification for not paying much attention to the second-order moment (i.e. $\mathrm{Cov}[\boldsymbol{x}_{t-\delta}|\boldsymbol{x}_t]$) is that, with enough, and small enough, denoising steps, the posterior covariance has a simple (diagonal) form available in closed-form (Anderson, 1982; Song et al., 2021). However, taking many small steps in an inherently sequential algorithm is not easily parallelized, such that a trade-off arises between sample quality and sampling time.

To speed up image generation, one can formulate diffusion in the (smaller) latent space of a pretrained image autoencoder (Rombach et al., 2022), distill the sampling process into a one-step network (Luo et al., 2023; Zhou et al., 2024a), or express the stochastic denoising process as an equivalent deterministic ODE that can be accelerated by appropriate choices of (e.g. higher-order) ODE solvers (Song et al., 2021; Lu et al., 2022; Karras et al., 2022; Zheng et al., 2023; Zhou et al., 2024b; Chen et al., 2024). Here, we follow an alternative line of recent research that has shown that standard DDPM sampling can be performed with fewer but larger steps when using a more accurate model of the posterior covariance (Bao et al., 2022b;a; Nichol & Dhariwal, 2021b; Rissanen et al., 2025). As it turns out, any score network that has been (well) trained to approximate the posterior mean contains all the information needed to estimate the covariance, too. This relationship has been formalized recently through a generalization of Tweedie's formula (Efron, 2011) to higher order

moments (Manor & Michaeli, 2021), revealing an analytical relation between high-order posterior moments and derivatives of the posterior mean (or, alternatively, of the 'score' function). This second-order information contained in pre-trained diffusion models can be distilled into parametric models of the covariance, either by differentiating through the score network exactly (Ou et al., 2024) or approximately (Manor & Michaeli, 2021), or by reformulating the posterior covariance as the minimum mean squared error (MSE) estimator of a quantity involving the posterior mean (Meng et al., 2021; see also Background). However, for models that generate color images with $D = d^2$ pixels, the full posterior covariance matrix (or, equivalently, its square root) has a large memory footprint $(3D \times 3D)$ implying $\mathcal{O}(D^2)$ sampling complexity, calling for more tractable approximations. This tradeoff is not unlike that encountered in neural network optimization, where accurately modeling the (second-order) curvature of the loss enables the use of larger learning rates, yet loss Hessians are large objects that can only be estimated in approximate, memory-efficient forms (Martens & Grosse, 2015; Garcia et al., 2023; Goldfarb et al., 2020). Rissanen et al. (2025) have recently leveraged this connection to improve image restoration.

All recent attempts at modeling denoising posterior covariances have assumed a diagonal or low-rank structure, which we argue is very restrictive. Here, we develop a new covariance model for image DDPMs (Fig. 1A) which accurately and efficiently captures the strong yet non-diagonal spatio-chromatic correlations between both neighbouring pixels and color channels present in natural images (Fig. 1B; Burton & Moorhead, 1987; Cui et al., 2020; Fairman & Brill, 2004). These chromatic and spatial correlations are approximately separable (Provenzi et al., 2016), and therefore Kronecker-factorizable, and the spatial component can be compactly represented in the frequency domain of the Discrete Cosine Transform (DCT) owing to approximate translation invariance (Hyvärinen et al., 2009). The resulting 'K-DCT' model is described in detail in Section 3.2; Fig. 1B (bottom) shows that it provides a good fit to the marginal (i.e. prior) CIFAR-10 covariance, and this paper explores its use for approximating the posterior covariances that arise in image denoising – an example of which is shown in Fig. 1C (top) along with its best K-DCT approximation (bottom). Starting from pre-trained score models, we learn K-DCT-structured amortizations of the (input-dependent) posterior covariance $\text{Cov}(\boldsymbol{x}_{t-\delta}|\boldsymbol{x}_t) \approx \text{K-DCT}(\boldsymbol{x}_t; \theta)$. On CIFAR-10, Celeb-A and ImageNet, we show that this leads to both better image generation (lower FID; Heusel et al., 2017) and better statistical models (lower negative log-likelihood) compared to previous diagonal approximations.

## 2 BACKGROUND

In this section, we begin by providing important background on the general Gaussian denoising problem, highlighting two ways of obtaining the mean and covariance of the denoising posterior distribution. We then discuss how these two denoising strategies can be applied to the sampling process in DDPMs, which we also summarize.

### 2.1 THE GAUSSIAN DENOISING PROBLEM

Consider a random vector $\boldsymbol{x} \in \mathbb{R}^n$ drawn from some distribution $q(\boldsymbol{x})$. Given a noisy observation $\tilde{\boldsymbol{x}} \sim q(\tilde{\boldsymbol{x}}|\boldsymbol{x}) = \mathcal{N}(\tilde{\boldsymbol{x}}; \boldsymbol{x}, \sigma^2 I)$, what can be said about $\boldsymbol{x}$? Whilst the posterior distribution $q(\boldsymbol{x}|\tilde{\boldsymbol{x}})$ is generally intractable (e.g. the prior $q(\boldsymbol{x})$ may not be Gaussian), there are at least two ways of obtaining its moments.

**Posterior moments via Tweedie's 1$^{\text{st}}$- and 2$^{\text{nd}}$-order formulae**  Posterior moments can be derived from the score function, $\nabla_{\tilde{\boldsymbol{x}}} \log \tilde{q}(\tilde{\boldsymbol{x}})$, where $\tilde{q}(\tilde{\boldsymbol{x}}) = \int d\boldsymbol{x}\, q(\tilde{\boldsymbol{x}}|\boldsymbol{x})\, q(\boldsymbol{x})$ is the marginal distribution of noisy observations. Tweedie's formula (Efron, 2011; Robbins, 1992) classically relates the posterior mean to the score function:

$$\mu^{\star}(\tilde{\boldsymbol{x}}) \triangleq \mathbb{E}[\boldsymbol{x}|\tilde{\boldsymbol{x}}] = \tilde{\boldsymbol{x}} + \sigma^2 \nabla_{\tilde{\boldsymbol{x}}} \log \tilde{q}(\tilde{\boldsymbol{x}}). \tag{1}$$

A similar relationship exists between the posterior covariance and the *second* derivative of $\log \tilde{q}(\cdot)$ (i.e. the Jacobian of the score; Manor & Michaeli, 2021; Meng et al., 2021):

$$\Sigma^{\star}(\tilde{\boldsymbol{x}}) \triangleq \text{Cov}[\boldsymbol{x}|\tilde{\boldsymbol{x}}] = \sigma^2 \nabla_{\tilde{\boldsymbol{x}}} \mu^{\star}(\tilde{\boldsymbol{x}}) = \sigma^2 \left( I + \sigma^2 \nabla_{\tilde{\boldsymbol{x}}}^2 \log \tilde{q}(\tilde{\boldsymbol{x}}) \right).$$

**Posterior moments as least-squares estimators**  When one has access to $(\boldsymbol{x}, \tilde{\boldsymbol{x}})$ pairs (e.g. via simulation: $\boldsymbol{x} \sim q(\boldsymbol{x})$, $\tilde{\boldsymbol{x}} \sim q(\tilde{\boldsymbol{x}}|\boldsymbol{x})$), one can estimate posterior moments from data by minimizing a

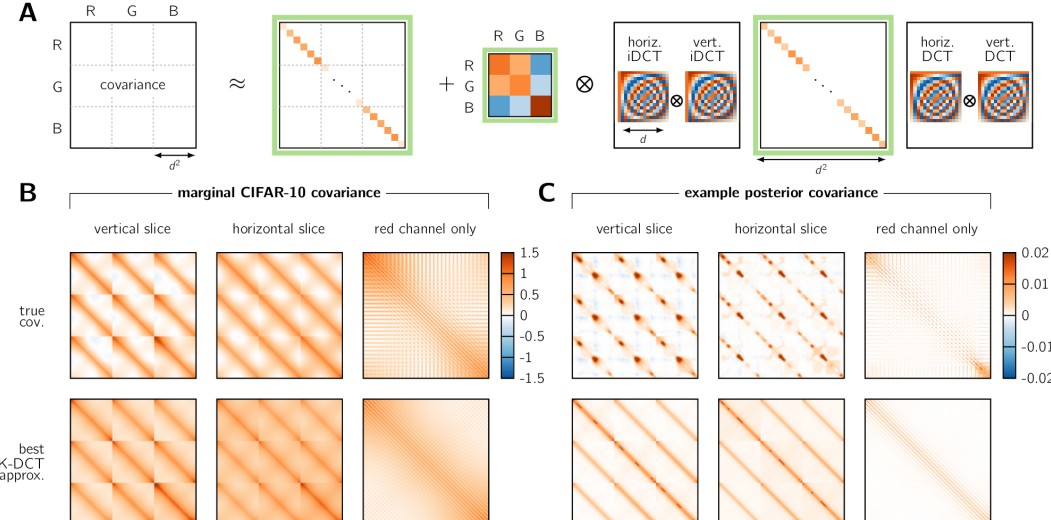

Figure 1: **Covariance matrices for denoising diffusion probabilistic models are well approximated by a Kronecker-DCT (K-DCT) structure.** (**A**) Illustration of our K-DCT covariance approximation (Eq. 12), with learnable parameters indicated in green. $\otimes$ denotes the Kronecker product. (**B**) Top: covariance of the CIFAR-10 dataset (image width $d = 32$), shown across RGB channels ($3 \times 3$ block structure) but restricted to pixels along the vertical (left) and horizontal (center) image midlines, and shown in full ($d \times d$ block structure, right) but for the red channel only. Bottom: same visualizations of the nearest (in minimum squared error sense) approximation of the CIFAR-10 covariance that conforms to the structure shown in (A). See also Figs. 4 and 5 for further analyses of how Eq. 12 accurately describes the marginal covariance structure of larger datasets (ImageNet and CelebA). (**C**) Same as (B), for an example posterior covariance matrix obtained from the Jacobian of a score network (see Eq. 2) pre-trained on CIFAR-10 (Bao et al., 2022a), evaluated at a partially denoised sample (600 denoising steps, i.e. roughly mid-way through denoising). See also Fig. 6 for a further dissection of how Eq. 12 performs at various stages of denoising on CIFAR-10.

squared error loss. Indeed, for any deterministic function $g(\cdot)$, the conditional expectation of $g(\boldsymbol{x})$ under the posterior $q(\boldsymbol{x}|\tilde{\boldsymbol{x}})$ is the solution to a mean-squared error minimization problem:

$$\mathbb{E}_{q(\boldsymbol{x}|\tilde{\boldsymbol{x}})}[g(\boldsymbol{x})] = \text{argmin}_{y(\cdot)} \, \mathbb{E}_{q(\tilde{\boldsymbol{x}}|\boldsymbol{x})q(\boldsymbol{x})} \left[ \|y(\tilde{\boldsymbol{x}}) - g(\boldsymbol{x})\|^2 \right]. \tag{2}$$

Thus, one can estimate the posterior mean function (i.e. $g(\boldsymbol{x}) = \boldsymbol{x}$), in parametric form $\mu_\theta(\tilde{\boldsymbol{x}})$, by minimizing

$$\theta^\star = \text{argmin}_\theta \, \mathbb{E}_{q(\tilde{\boldsymbol{x}}|\boldsymbol{x})q(\boldsymbol{x})} \left[ \|\mu_\theta(\tilde{\boldsymbol{x}}) - \boldsymbol{x}\|^2 \right]. \tag{3}$$

with the expectation typically estimated via Monte-Carlo sampling of $(\boldsymbol{x}, \tilde{\boldsymbol{x}})$ pairs. Similarly, a parametric posterior covariance model $\Sigma_\theta(\tilde{\boldsymbol{x}})$ can be learned by minimizing

$$\theta^\star = \text{argmin}_\theta \, \mathbb{E}_{q(\tilde{\boldsymbol{x}}|\boldsymbol{x})q(\boldsymbol{x})} \left[ \left\| \Sigma_\theta(\tilde{\boldsymbol{x}}) - (\boldsymbol{x} - \boldsymbol{\mu}^\star(\tilde{\boldsymbol{x}}))(\boldsymbol{x} - \boldsymbol{\mu}^\star(\tilde{\boldsymbol{x}}))^\top \right\|^2 \right] \tag{4}$$

with $\boldsymbol{\mu}^\star$ either derived from the score function through Eq. 1, or parametrically estimated using Eq. 3.

## 2.2 DENOISING DIFFUSION PROBABILISTIC MODELS

DDPMs (Ho et al., 2020) are probabilistic models that allow sampling from an arbitrary distribution $q(\boldsymbol{x}_0)$. Just as in the Gaussian denoising problem discussed above, DDPMs define a whole collection of noisy observations $\{\boldsymbol{x}_1, \ldots, \boldsymbol{x}_T\}$, obtained by sequentially down-scaling, and adding Gaussian noise to, each data sample $\boldsymbol{x}_0$: $\boldsymbol{x}_t = \sqrt{\alpha_t}\boldsymbol{x}_{t-1} + \sqrt{\beta_t}\boldsymbol{\epsilon}_t$ with $\boldsymbol{\epsilon}_t \sim \mathcal{N}(0, I)$, where $\beta_t$ is a time-dependent diffusion coefficient also known as the 'noising schedule'. Typically, $\alpha_t = 1 - \beta_t$, a choice that preserves the total variance of $\boldsymbol{x}_t$ at each step $t$. Having introduced this forward 'noising' Markov chain, the data distribution can be expressed as $q(\boldsymbol{x}_0) = \int d\{\boldsymbol{x}_1, \ldots, \boldsymbol{x}_T\} q(\boldsymbol{x}_T) \prod_{t=1}^{T} q(\boldsymbol{x}_{t-1}|\boldsymbol{x}_t)$. Therefore, sampling from $q(\boldsymbol{x}_0)$ can be achieved by sampling from $q(\boldsymbol{x}_T)$ and then running a sequence

of small denoising steps whereby each $\boldsymbol{x}_{t-1}$ is obtained from $\boldsymbol{x}_t$ by sampling the relevant denoising posterior $q(\boldsymbol{x}_{t-1}|\boldsymbol{x}_t)$. Critically, for a sufficiently long noising process, $\boldsymbol{x}_T$ is approximately normally distributed and is therefore trivial to sample.

In general, each posterior $q(\boldsymbol{x}_{t-1}|\boldsymbol{x}_t)$ is intractable, and is normally approximated by a Gaussian:

$$q(\boldsymbol{x}_{t-1}|\boldsymbol{x}_t) \approx \mathcal{N}\left(\boldsymbol{x}_{t-1}; \boldsymbol{\mu}_{t-1}(\boldsymbol{x}_t), \Sigma_{t-1}(\boldsymbol{x}_t)\right). \tag{5}$$

Note that it is possible to generate a sample $\boldsymbol{x}_0$ using a number of denoising steps smaller than $T$, by merging any number of consecutive denoising steps into a single one ('skip-step DDPM'). This is done by leveraging the fact that, under the noising model, any $\boldsymbol{x}_t$ is a linear-Gaussian transformation not only of $\boldsymbol{x}_{t-1}$ but of any previous $\boldsymbol{x}_{s<t}$. This leads to simple affine relationships between $\{\boldsymbol{\mu}_{t-1}(\boldsymbol{x}_t), \Sigma_{t-1}(\boldsymbol{x}_t)\}$ and the more general $\{\boldsymbol{\mu}_s(\boldsymbol{x}_t), \Sigma_s(\boldsymbol{x}_t)\}$ (Appendix B). It is precisely when skipping steps that the posterior covariance becomes less diagonally dominant, such that it becomes important to accurately model its structure – the focus of this paper. We now discuss how estimates of $\boldsymbol{\mu}_{t-1}(\boldsymbol{x}_t)$ and $\Sigma_{t-1}(\boldsymbol{x}_t)$ can be obtained.

**Posterior mean**    The posterior mean function $\boldsymbol{\mu}_{t-1}(x_t)$ is typically obtained indirectly by estimating the *effective noise* term $\boldsymbol{\epsilon}_t \triangleq \frac{\boldsymbol{x}_t - \sqrt{\bar{\alpha}_t}\boldsymbol{x}_0}{\sqrt{\bar{\beta}_t}}$ that transformed $\boldsymbol{x}_0$ into $\boldsymbol{x}_t$, using standard notation $\bar{\alpha}_t \triangleq \prod_{s=0}^t \alpha_s$ and $\bar{\beta}_t \triangleq 1 - \bar{\alpha}_t$. Indeed, simple affine transformations exist between the conditional expectation $\mathbb{E}[\boldsymbol{\epsilon}_t|\boldsymbol{x}_t]$ and $\mathbb{E}[\boldsymbol{x}_0|\boldsymbol{x}_t]$, and further towards $\boldsymbol{\mu}_{t-1}(\boldsymbol{x}_t) \equiv \mathbb{E}[\boldsymbol{x}_{t-1}|\boldsymbol{x}_t]$ as follow:

$$\mathbb{E}[\boldsymbol{x}_0|\boldsymbol{x}_t] = \frac{\boldsymbol{x}_t - \sqrt{\bar{\beta}_t}\mathbb{E}[\boldsymbol{\epsilon}_t|\boldsymbol{x}_t]}{\sqrt{\bar{\alpha}_t}}, \quad \boldsymbol{\mu}_{t-1}(\boldsymbol{x}_t) = \frac{\sqrt{\bar{\alpha}_{t-1}}\,\beta_t\mathbb{E}[\boldsymbol{x}_0|\boldsymbol{x}_t] + \sqrt{\alpha_t}\bar{\beta}_{t-1}\boldsymbol{x}_t}{\bar{\beta}_t}. \tag{6}$$

In practice, a neural network $\boldsymbol{\epsilon}_\theta(\boldsymbol{x}_t, t)$ is trained to approximate $\mathbb{E}[\boldsymbol{\epsilon}_t|\boldsymbol{x}_t]$, and used to evaluate Eq. 6 (see also Appendix G.1 for more practical details, including the image-specific use of clipping).

**Posterior covariance**    The posterior covariance function, $\Sigma_{t-1}(\boldsymbol{x}_t)$, is often approximated by one of two time-dependent, but $\boldsymbol{x}_t$-independent, heuristics: $\beta_t I$ ('large'), or $\tilde{\beta}_t I$ ('small') with $\tilde{\beta}_t \triangleq \frac{1-\bar{\alpha}_{t-1}}{1-\bar{\alpha}_t}\beta_t$. These become equal, and exact, in the limit of many small (de-)noising steps, i.e. a limit where $\boldsymbol{x}_0$ contains much less information about $\boldsymbol{x}_{t-1}$ than does $\boldsymbol{x}_t$. For realistically small horizons $T$, however, the posterior covariance may be far from being a scalar, or even a diagonal matrix (e.g. Fig. 1C). Several works have sought to learn better models of the posterior covariance in parametric form. Similarly to the posterior mean function, the posterior covariance function $\Sigma_{t-1}(\boldsymbol{x}_t)$ is mathematically related to the covariance of the noise, $\text{Cov}[\boldsymbol{\epsilon}_t|\boldsymbol{x}_t]$, as follows:

$$\text{Cov}[\boldsymbol{x}_0|\boldsymbol{x}_t] = \frac{\bar{\beta}_t}{\bar{\alpha}_t}\text{Cov}[\boldsymbol{\epsilon}_t|\boldsymbol{x}_t], \;\; \Sigma_{t-1}(\boldsymbol{x}_t) = \tilde{\beta}_t I + \frac{\beta_t^2\bar{\alpha}_{t-1}}{(1-\bar{\alpha}_t)^2}\text{Cov}[\boldsymbol{x}_0|\boldsymbol{x}_t]. \tag{7}$$

Hence, to perform the denoising sampling step of Eq. 5 given a pretrained model that already approximates the posterior mean of the effective noise term, it is sufficient to learn a parametric model $\mathcal{E}_\phi(\boldsymbol{\epsilon}_t, t)$ of $\text{Cov}(\boldsymbol{\epsilon}_t|\boldsymbol{x}_t)$. This model needs to have a manageable memory footprint, and its matrix square root (required for sampling) must afford computationally tractable matrix-vector products. The K-DCT covariance model we propose here is equally applicable to the two ways of obtaining posterior covariances described in Section 2.1: either via derivatives of the score function (Tweedie's $2^{\text{nd}}$-order formula), or via direct least-squares estimation from data. In the following, we describe their specific application to the DDPM denoising posterior.

### 2.3    Learning posterior covariance approximations for DDPMs

**Score derivative-based approach**    Leveraging the connection between the denoising posterior covariance and the Jacobian of the score (Manor & Michaeli, 2021), Ou et al. (2024) learned a parametric diagonal covariance model $\mathcal{E}_\phi(\boldsymbol{x}_t, t) = \text{diag}(\boldsymbol{\varepsilon}_\phi(\boldsymbol{x}_t, t))$ by minimizing the following 'optimal covariance matching' (OCM) objective:

$$\mathcal{L}_{\text{OCM}}(\phi) = \mathbb{E}_{q(\boldsymbol{x}_0)q(\boldsymbol{x}_t|\boldsymbol{x}_0)} \left\| \boldsymbol{\varepsilon}_\phi(\boldsymbol{x}_t, t) - \text{diag}\left(I - \sqrt{\bar{\beta}_t}\nabla_{\boldsymbol{x}_t}\boldsymbol{\epsilon}_\theta(\boldsymbol{x}_t, t)\right) \right\|_2^2, \tag{8}$$

where $\epsilon_\theta(\cdot, \cdot)$ is a pretrained first-order model approximating $\epsilon_t$ (see Section 2.2), and $\mathrm{diag}(M)$ extracts the diagonal of matrix $M$. Here, we will adapt the objective of Eq. 8, which targets the gradient of the score, to an equivalent formulation which targets $\mathrm{Cov}(\epsilon_t|\boldsymbol{x}_t)$ instead (Eqs. 2 and 7). For non-diagonal approximations (such as ours; see below), one cannot afford materializing the residual in Eq. 8 in order to compute its squared norm. To circumvent this, Ou et al. used an unbiased stochastic estimator of the corresponding gradient, obtained by automatically differentiating through the following surrogate objective

$$\tilde{\mathcal{L}}_{\mathrm{OCM}}(\phi) = \mathbb{E}_{q(\boldsymbol{x}_0)q(\boldsymbol{x}_t|\boldsymbol{x}_0)}\mathbb{E}_{\boldsymbol{v}\sim p(\boldsymbol{v})} \left\| \boldsymbol{\varepsilon}_\phi(\boldsymbol{x}_t, t) - \boldsymbol{v} \odot \left( \boldsymbol{v} - \sqrt{\bar{\beta}_t}\nabla_{\boldsymbol{x}_t}\boldsymbol{\epsilon}_\theta(\boldsymbol{x}_t, t)\boldsymbol{v} \right) \right\|_2^2, \quad (9)$$

where $\odot$ denotes the Hadamard (element-wise) product. The inner expectation can be stochastically estimated via Monte-Carlo sampling of $\boldsymbol{v}$ from an isotropic Rademacher distribution. Note that the $\nabla_{\boldsymbol{x}_t}\boldsymbol{\epsilon}_\theta(\boldsymbol{x}_t, t)\boldsymbol{v}$ term is a Jacobian-vector product (JVP) that can be calculated efficiently using forward-mode auto-differentiation (AD). As it does not depend on $\phi$, this JVP needs not be further differentiated (i.e. no need for nested AD). Here, we will adapt this approach to deal with more general, non-diagonal covariance models (Appendices E and F).

**MMSE approach** Meng et al. (2021) leveraged the least-squares estimator interpretation of the denoising posterior moments (Section 2.1) to learn amortizations of higher-order derivatives of any data (log) distribution. In turn, they showed that a good second-order score approximation leads to better denoising uncertainty quantification. More recently, Bao et al. (2022a) followed a similar approach to fit a parametric model of $\mathrm{Cov}(\epsilon_t|\boldsymbol{x}_t)$ in the form $\mathcal{E}_\phi(\boldsymbol{x}_t, t) = \mathrm{diag}(\boldsymbol{\varepsilon}_\phi(\boldsymbol{x}_t, t))$, using a MMSE objective. Given independent $(\boldsymbol{x}_0, \boldsymbol{\epsilon}_t)$ pairs and the associated $\boldsymbol{x}_t = \sqrt{\bar{\alpha}_t}\boldsymbol{x}_0 + \sqrt{\bar{\beta}_t}\boldsymbol{\epsilon}_t$, their 'noise prediction residual' (NPR) objective reads

$$\mathcal{L}_{\mathrm{NPR}}(\phi) = \mathbb{E}_{\boldsymbol{x}_0\sim q(\boldsymbol{x}_0); \boldsymbol{\epsilon}_t\sim\mathcal{N}(0,I)} \left\| \boldsymbol{\varepsilon}_\phi(\boldsymbol{x}_t, t) - (\boldsymbol{\epsilon}_t - \boldsymbol{\epsilon}_\theta(\boldsymbol{x}_t, t))^2 \right\|_2^2. \quad (10)$$

This objective again relies on a pretrained first-order noise predictor $\epsilon_\theta(\boldsymbol{x}_t, t)$. Similar to the OCM objective, we will adapt the NPR objective to deal with more general, non-diagonal covariance models.

## 3 COVARIANCE PARAMETERIZATIONS

Tractable evaluation and differentiation of both the OCM (Eq. 8) and NPR (Eq. 10) objectives places constraints on the form of covariance approximation ($\mathcal{E}_\phi(\cdot)$) that may be used. One highly flexible, but also highly intractable, choice would be to parameterize the Cholesky factor of the entire $3D \times 3D$ covariance matrix, where $D = d^2$ is the number of image pixels – this leads to a prohibitive $\mathcal{O}(D^2)$ memory and compute complexity. In this work, we introduce a model $\mathcal{E}_\phi$ that provides not only a better inductive bias for image generation than previous proposals (briefly reviewed below), whilst affording efficient training and sampling.

### 3.1 EXISTING PARAMETERIZATIONS

As previously mentioned, a popular covariance approximation is the diagonal parameterization: e.g. Nichol & Dhariwal (2021a) and Ou et al. (2024) parameterize $\boldsymbol{\varepsilon}_\phi(\boldsymbol{x}_t, t) \in \mathbb{R}^{3D}$ such that $\mathcal{E}_\phi^{\mathrm{diag}}(\boldsymbol{x}_t, t) = \mathrm{diag}(\boldsymbol{\varepsilon}_\phi(\boldsymbol{x}_t, t)) \in \mathbb{R}^{3D \times 3D}$. This has $\mathcal{O}(D)$ (linear) complexity in both training and sampling, but fails to take into account pairwise correlations between pixels. To capture the dominant patterns of pairwise correlations under the denoising posterior, Meng et al. (2021) added a low-rank component to the diagonal, resulting in:

$$\mathcal{E}_\phi(\boldsymbol{x}_t, t) = \mathrm{diag}(\boldsymbol{\varepsilon}_\phi(\boldsymbol{x}_t, t)) + R_\phi(\boldsymbol{x}_t, t)R_\phi(\boldsymbol{x}_t, t)^\top \quad (11)$$

where $R_\phi \in \mathbb{R}^{3D \times r}$ with $r \ll 3D$. However, we find that the eigenvalue spectra of image denoising posterior covariances tend to decay slowly (see also Van der Schaaf & van Hateren, 1996), such that $r$ might need to be fairly large to capture useful structure. This is corroborated by Meng et al.'s qualitative results on the MNIST dataset, where they used $r = 50$ (i.e. 6% of $D$) to obtain a posterior covariance approximation that contained the patterns of denoising uncertainty between digits that

one would intuitively expect. In the CIFAR-10 example of Fig. 1C, capturing 90% of the variance in the denoising covariance matrix requires setting $r = 635 \approx 20\%$ of $3D$. For larger images, such as those in the datasets we consider here (CelebA and ImageNet), better forms of approximation are needed that can capture the full rank of the posterior covariance without introducing an additional compute/memory tradeoff.

### 3.2 PROPOSED K-DCT PARAMETERIZATION

Motivated by key statistical properties of natural images (recall Introduction and Fig. 1), we propose the following covariance model for image DDPMs:

$$\mathcal{E}_\phi(\boldsymbol{x}_t, t) = \mathrm{diag}(\boldsymbol{\varepsilon}_\phi(\boldsymbol{x}_t, t)) + \underbrace{C_\phi(\boldsymbol{x}_t, t) C_\phi(\boldsymbol{x}_t, t)^\top}_{\text{inter-channel}} \otimes \underbrace{(\overbrace{F^\top}^{\text{horiz.}} \otimes \overbrace{F^\top}^{\text{vert.}}) \mathrm{diag}(\boldsymbol{\lambda}_\phi(\boldsymbol{x}_t, t)) (F \otimes F)}_{\text{inter-pixel}}. \quad (12)$$

As detailed below, this model is grounded both in the empirical structure of image covariances described in previous literature and seen in our own observations on CIFAR-10, and in the spectral theory of discrete cosine transforms. The first (diagonal) term absorbs any diagonal contribution that the second (Kronecker-DCT) term might not capture, thereby ensuring that the model is at least as expressive as previous diagonal models we compare to. The second term models correlations between colored pixels as the Kronecker product ($\otimes$) of inter-channel (color) correlations ($C_\phi C_\phi^\top$ term) and inter-pixel (spatial) correlations; in other words, we capitalize on the approximately *separable* spatio-chromatic correlation structure of natural images, whereby e.g. the red and blue content of two pixels are correlated in the same way irrespective of where these two pixels are located (Provenzi et al., 2016). For the spatial component, we reason that natural images – seen as continuous functions of the infinite plane – have an approximately translation invariant distribution; thus, their covariance operator has the Fourier modes as eigenfunctions (Hyvärinen et al., 2009). Therefore, for discretized (finite-size) images, the first practical parameterization that comes to mind is a diagonal matrix in the orthonormal basis given by the Discrete Fourier Transform (DFT) matrix. However, the DFT assumes cyclic boundary conditions which natural bounded images do not have – in other words, their covariance is not circulant (as a DFT-based model would assume) but rather Toeplitz (Rissanen et al., 2025). In fact, empirically we find that the CIFAR-10 marginal covariance, in both the horizontal and vertical image directions, is the superposition of a Toeplitz component and a Hankel ("90-deg rotated Toeplitz") component (Fig. 1B). This suggests using the discrete cosine transform (DCT) instead: matrices diagonalized by the DCT have indeed been shown theoretically to possess precisely this Toeplitz + Hankel structure (Sanchez et al., 2002). We therefore consider a spatial covariance diagonalized by the 2-dimensional DCT operator $F \otimes F$, which applies the standard 1-dimensional DCT operator $F \in \mathbb{R}^{d \times d}$ (Strang, 1999) to both the vertical and horizontal image dimensions (note that $F^{-1} = F^\top$). The $D$ eigenvalues are parameterized in positive real form as $\boldsymbol{\lambda}_\phi(\boldsymbol{x}_t, t)$. Visual intuition for the expressiveness of this spatial covariance parameterization is given in Fig. 3.

This model has a small, $\mathcal{O}(D)$ memory footprint (i.e. the size of a single image). In the next two subsections, we show that it is also amenable to efficient, $\mathcal{O}(D \log d)$ training and sampling.

**Efficient training**  The first step in learning non-diagonal covariance models using the OCM or NPR objectives is to extend their diagonal covariance formulations (Eqs. 8 and 10) to the more general, non-diagonal case. The corresponding expressions are provided in Eqs. 29 and 40 (Appendix F). Our parametrization in Eq. 12 enables efficient evaluation and differentiation of these objectives, much cheaper than the naive approach ($\mathcal{O}(D \log d)$ instead of $\mathcal{O}(D^2)$ compute complexity). Indeed, the two most expensive operations are matrix-vector products with $\mathcal{E}_\phi$, and the squared Frobenius norm $\|\mathcal{E}_\phi\|_F^2$ – both of which can be computed efficiently due to the tensor product structure of Eq. 12. In particular, for an image $V \in \mathbb{R}^{3 \times d \times d}$, the corresponding matrix-vector product can be computed as follows:

$$(\mathcal{E}_\phi(\boldsymbol{x}_t, t)\mathrm{vec}(V))^{cij} = \varepsilon^{cij} V^{cij} + (F^\top)^j_n (F^\top)^i_m \left( \boldsymbol{\lambda}^{mn} F^n_q F^m_p \overbrace{V^{epq} \underbrace{(CC^\top)^c_e}_{\mathcal{O}(3^2)}}^{\mathcal{O}(D \log d)} \right) \quad (13)$$

where $\varepsilon \equiv \boldsymbol{\varepsilon}_\phi(\boldsymbol{x}_t, t)$, $\boldsymbol{\lambda} \equiv \boldsymbol{\lambda}_\phi(\boldsymbol{x}_t, t)$ and $C \equiv C_\phi(\boldsymbol{x}_t, t)$. In Eq. 13, $\mathrm{vec}(\cdot)$ is the tensor vectorization operation, and we have used standard Einstein notation whereby repeated indices occurring in

opposite super-/sub-scripts are summed over. In the context of our training losses, $V$ may be either $\epsilon_t$, $\epsilon_\theta(\boldsymbol{x}_t, t)$, or Rademacher samples used in stochastic trace estimators. The above equation allows us to never explicitly construct large $3D \times 3D$ matrices, but instead only perform element-wise multiplication and linear transformations of dimension $d$. Note that in theory, products with the DCT matrix such as $F_p^m V^p$ can be computed in $\mathcal{O}(d \log d)$ complexity, in practice we find that GPU-accelerated matrix multiplications with $F$ are faster. Pseudocode for efficiently evaluating the training loss is given in Algorithm 2 (NPR) and Algorithm 3 (OCM).

**Efficient sampling**   While reducing training complexity is desirable, using the covariance model for image generation also requires efficient sampling. Sampling from the denoising posterior (Eq. 5) or the skip-step posterior (Eqs. 16 and 17) is traditionally done by multiplying a random normal vector by the matrix square-root of $\mathcal{E}_\phi(\boldsymbol{x}_t, t)$. While the latter is difficult to obtain for our proposed parameterization due to the sum in Eq. 12, we can instead sample and add two independent samples from the two corresponding multivariate normal distributions. For the first (diagonal) term, this is straightforward. The second term admits a simple matrix square root, $C \otimes \big((F^\top \otimes F^\top)\mathrm{diag}\big(\boldsymbol{\lambda}^{1/2}\big)\big)$, such that sampling from the corresponding Gaussian can be written analogously to Eq. 13 as:

$$(F^\top)^j{}_n (F^\top)^i{}_m \bigg( \Big(\boldsymbol{\lambda}^{\frac{1}{2}}\Big)^{mn} \xi^{emn} C^c{}_e \bigg) \tag{14}$$

where $\xi \in \mathbb{R}^{3D}$ is a standard random normal vector. Pseudocode for sampling is given in Algorithm 1; note that the covariance $\Sigma_{t-1}(\boldsymbol{x}_t)$ from which we must sample is not exactly the same as the covariance of the noise ($\mathcal{E}_\phi$) for which pseudocode is given, but it has the same structure (c.f. Eq. 7).

Empirical comparison of training and sampling time cost between diagonal covariance and our K-DCT model is provided in Table 6, which shows little computation overhead for our parameterization.

---

**Algorithm 1** Sampling from $\mathcal{E}_\phi(\boldsymbol{x}_t, t)$

---

**Require:** Covariance model components $\{\boldsymbol{\varepsilon}_\phi, C_\phi, \boldsymbol{d}_\phi\}$ with trained parameter set $\phi$, partially denoised $\boldsymbol{x}_t$ at a given $t$, two independent Gaussian samples $\boldsymbol{\xi}_1, \boldsymbol{\xi}_2 \sim \mathcal{N}(0, I)$.
**Ensure:** $\boldsymbol{g}$ is a sample from $\mathcal{N}\big(0, \mathcal{E}_\phi(\boldsymbol{x}_t, t)\big)$
 1: Compute model outputs $\boldsymbol{\varepsilon} \leftarrow \boldsymbol{\varepsilon}_\phi(\boldsymbol{x}_t, t)$, $C \leftarrow C_\phi(\boldsymbol{x}_t, t)$ and $\boldsymbol{d} \leftarrow \boldsymbol{d}_\phi(\boldsymbol{x}_t, t)$
 2: Compute $\tilde{\boldsymbol{\xi}}_1 \leftarrow \boldsymbol{\varepsilon}^{\frac{1}{2}} \odot \boldsymbol{\xi}_1$   # diagonal part; $\odot$ denotes the element-wise product
 3: Compute $\tilde{\boldsymbol{\xi}}_2 \leftarrow$ 2D-iDCT(einsum($\boldsymbol{d}^{\frac{1}{2}}, C, \boldsymbol{\xi}_2$, `'ij,ck,kij->cij'`))   # non-diagonal part
 4: Compute $\boldsymbol{g} \leftarrow \tilde{\boldsymbol{\xi}}_1 + \tilde{\boldsymbol{\xi}}_2$

---

# 4   EXPERIMENTS & RESULTS

In this section, we run experiments to validate our hypothesis that the K-DCT covariance model (Eq. 12) provides a better inductive bias for image DDPMs, leading to better generative models. We use previously published, first-order UNet models pre-trained on various datasets (Table 4), add additional UNet heads to decode the various terms of our covariance model (details below), and optimize the parameters of these new heads w.r.t. the generalized NPR or OCM objectives. We systematically compare our approach with results previously reported for the same first-order models but with diagonal covariance approximations[1]. Our results are summarized in Fig. 2 and Table 2.

**Model structure and training**   We use the same parameter sharing strategy as used in Bao et al. (2022a); Ou et al. (2024), where the (pretrained) first-order noise predictor $\epsilon_\theta$ and the covariance model $\mathcal{E}_\phi$ share most of their parameters, as follows:

$$\epsilon_\theta(\boldsymbol{x}_t, t) = \mathrm{NN}_1(\mathrm{UNet}(\boldsymbol{x}_t, t; \theta_1), \theta_2), \quad \mathcal{E}_\phi(\boldsymbol{x}_t, t) = \mathrm{NN}_2(\mathrm{UNet}(\boldsymbol{x}_t, t; \theta_1), \phi) \tag{15}$$

Here, $\theta_1$ and $\theta_2$ are *fixed* parameters of the pretrained model, and $\mathrm{NN}_2(\cdot; \phi)$ is a model that outputs the three key components of Eq. 12 ($\boldsymbol{\varepsilon}_\phi, C_\phi, \boldsymbol{d}_\phi$) and which we train using the *same* dataset and noising schedule as were used to pretrain the first-order model. In more detail, $\boldsymbol{\varepsilon}_\phi$ receives input

---

[1]Our code is built on previous work released by (Ou et al., 2024; Bao et al., 2022a) for fair comparison. The code is given in the supplementary material.

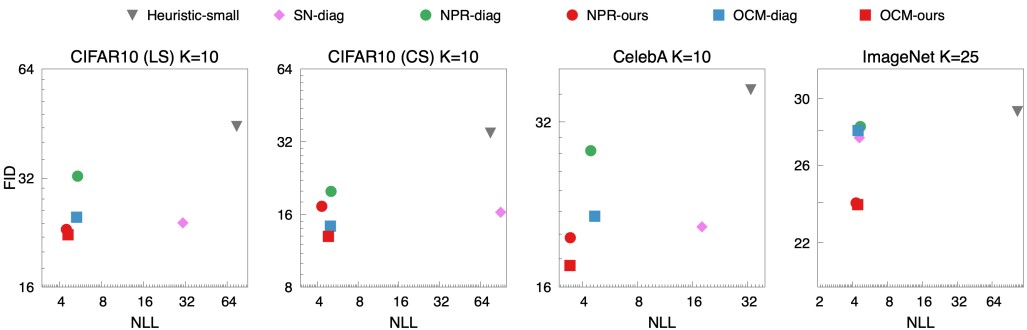

Figure 2: FID (log-scale) vs. NLL (log-scale) for different training objectives (OCM, NPR and SN) and different covariance models (heuristic, diagonal, ours) in the skip-step regime of few sampling steps ($K$). Our model consistently achieves both lower FID and NLL.

from the last *up*-block layer in the UNet, while $C_\phi$ and $d_\phi$ receive input from the last *middle*-block layer. Indeed, we reasoned that the former might require pixel-level information, while the latter two might benefit from more abstract features. Thus, our parameterization only requires training a smaller neural network compared to the UNet. Moreover, when compared with diagonal covariance models ($\varepsilon_\phi$ only), our non-diagonal model only requires the addition of two smaller components ($C_\phi$ and $d_\phi$) which adds negligible overhead. Refer to Table 5 for detailed model architectures.

**Datasets & compared methods**  Following the experimental setting of Bao et al. (2022a), we evaluate our full covariance model across several datasets and associated pre-trained first-order score networks: CIFAR10 (Krizhevsky et al., 2009) with linear (LS; Ho et al., 2020) and cosine (CS; (Nichol & Dhariwal, 2021a)) noising schedules, CelebA (Liu et al., 2015), and down-sampled ImageNet ($64 \times 64$; Deng et al., 2009. We borrow most of the implementation details and hyperparameter from Bao et al. (2022a) and compare our results with those previously reported for constant, diagonal heuristic covariance and for $x_t$-dependent diagonal covariance, summarized in Table 1.

**Evaluation**  For evaluation, we focus on two key metrics: statistical goodness of fit as measured by the average negative log-likelihood (NLL; $-\log p(x)$) of test images, and perceptual quality of generated images as measured by the FID (Heusel et al., 2017). We approximate the NLL via the standard evidence lower-bound (Ho et al., 2020), $-\log q(x) \leq \mathbb{E}_q \log \frac{p(x_{0:T:k})}{q(x_{1:T:k}|x_0)} \equiv -L_{\text{ELBO}}(x)$, where $p(\cdot)$ denotes the Markov denoising process, and $k$ denotes the number of skipped steps. The lower bound becomes tighter with more sampling steps, i.e. with smaller $k$. Here we have conservatively evaluated all inverses and log determinants involved in the ELBO using direct Cholesky decompositions, but see Fig. 8 for an efficient stochastic estimator that is applicable at scale (Appendix G.1). As shown in the lower block of Table 2, our K-DCT model consistently outperforms diagonal models in terms of NLL, regardless of the training objective used, be it score derivative-based (OCM) or MMSE-based

Table 1: Different covariance estimation methods ranked by increasing expressiveness

| Covariance | Type | Intuition |
|---|---|---|
| Heuristic large $\beta_t$ (Ho et al., 2020) | Isotropic constant | Cov. of $q(x_t|x_{t-1})$ |
| Heuristic small $\tilde{\beta}_t$ (Ho et al., 2020) | Isotropic constant | Cov. of $q(x_t|x_{t-1}, x_0)$ |
| SN-diagonal (Bao et al., 2022a) | Diagonal $x_t$-dependent | Learn from data, $\mathbb{E}(\epsilon_t^2|x_t)$ |
| NPR-diagonal (Bao et al., 2022a) | Diagonal $x_t$-dependent | Learn from data, $\text{Cov}(\epsilon_t|x_t)$ |
| OCM-diagonal (Ou et al., 2024) | Diagonal $x_t$-dependent | Learn from score, $\nabla_{x_t} \log \tilde{q}(x_t, t)$ |
| LowRank (Meng et al., 2021) | Low-rank $x_t$-dependent | Learn from data, $\text{Cov}(\epsilon_t|x_t)$ |
| NPR-K-DCT (Ours) | Full $x_t$-dependent | Learn from data, $\text{Cov}(\epsilon_t|x_t)$ |
| OCM-K-DCT (Ours) | Full $x_t$-dependent | Learn from score, $\nabla_{x_t} \log \tilde{q}(x_t, t)$ |

Table 2: FID score and NLL across various datasets with different sampling steps. Colors denote **1**[st] and **2**[nd] best (i.e. lowest) FID and NLL values.

| FID | CIFAR10 (LS) | | | CIFAR10 (CS) | | | CelebA $64 \times 64$ | | | ImageNet $64 \times 64$ | | |
|---|---|---|---|---|---|---|---|---|---|---|---|---|
| # Timesteps $K$ | 10 | 25 | 50 | 10 | 25 | 50 | 10 | 25 | 50 | 25 | 50 | 100 |
| Heuristic $\tilde{\beta}_t$ | 44.45 | 21.83 | 15.21 | 34.76 | 16.18 | 11.11 | 36.69 | 24.46 | 18.96 | 29.21 | 21.71 | 19.12 |
| Heuristic $\beta_t$ | 233.41 | 125.05 | 66.28 | 205.31 | 84.71 | 37.35 | 294.79 | 115.69 | 53.39 | 170.28 | 83.86 | 45.04 |
| SN-diagonal | 24.06 | 6.91 | 4.63 | 16.33 | 6.05 | 4.17 | 20.60 | 12.00 | 7.88 | 27.58 | 20.74 | 18.04 |
| NPR-diagonal | 32.35 | 10.55 | 6.18 | 19.94 | 7.99 | 5.31 | 28.37 | 15.74 | 10.89 | 28.27 | 20.89 | 18.06 |
| NPR-K-DCT (ours) | 23.06 | 9.12 | 5.92 | 17.26 | 7.79 | 5.58 | 19.69 | 13.72 | 9.80 | 23.98 | 18.90 | 17.44 |
| OCM-diagonal | 24.94 | 9.19 | 5.95 | 14.32 | 5.54 | 4.10 | 21.55 | 12.71 | 9.24 | 28.02 | 20.81 | 17.98 |
| OCM-K-DCT (ours) | 22.30 | 9.10 | 5.92 | 12.96 | 6.28 | 5.05 | 17.51 | 10.88 | 7.57 | 23.88 | 18.78 | 17.39 |
| **NLL $\approx$ -ELBO** | CIFAR10 (LS) | | | CIFAR10 (CS) | | | CelebA $64 \times 64$ | | | ImageNet $64 \times 64$ | | |
| # Timesteps $K$ | 10 | 25 | 50 | 10 | 25 | 50 | 10 | 25 | 50 | 25 | 50 | 100 |
| Heuristic $\tilde{\beta}_t$ | 74.95 | 24.98 | 12.01 | 75.96 | 24.94 | 11.96 | 33.42 | 13.09 | 7.14 | 105.87 | 46.25 | 22.02 |
| Heuristic $\beta_t$ | 6.99 | 6.11 | 5.44 | 6.51 | 5.55 | 4.92 | 6.67 | 5.72 | 4.98 | 5.81 | 5.20 | 4.70 |
| SN-diagonal | 30.79 | 11.83 | 7.13 | 90.85 | 19.81 | 9.72 | 18.09 | 8.05 | 5.29 | 4.56 | 4.18 | 3.95 |
| NPR-diagonal | 5.40 | 4.64 | 4.25 | 5.03 | 4.33 | 3.99 | 4.46 | 3.78 | 3.40 | 4.66 | 4.22 | 3.96 |
| NPR-K-DCT (ours) | 4.50 | 4.10 | 3.90 | 4.31 | 3.98 | 3.87 | 3.45 | 3.17 | 2.99 | 4.29 | 4.07 | 3.91 |
| OCM-diagonal | 5.32 | 4.63 | 4.25 | 4.99 | 4.34 | 3.99 | 4.69 | 3.86 | 3.43 | 4.45 | 4.15 | 3.93 |
| OCM-K-DCT (ours) | 4.61 | 4.17 | 4.02 | 4.82 | 4.24 | 3.94 | 3.44 | 3.16 | 2.99 | 4.41 | 4.14 | 3.93 |

(NPR). As expected, improvements are more significant in the case of fewer sampling steps where a diagonal covariance no longer provides a good approximation.

These improvements in NLL are largely reflected in FID improvements (see upper block of Table 2 where FIDs were evaluated based on 50k generated samples), especially in more aggressive skip-step regimes (lower $K$) and for larger images. Although FID and NLL are known to be somewhat loosely related (e.g. the 'squared-noise' (SN) diagonal approximation tends to do well in terms of FID but poorly on likelihoods; Bao et al., 2022a), overall our K-DCT model exhibits the best tradeoff between these two evaluation metrics amongst all models (Fig. 2). Example samples generated by our method can be found in Appendix I.

Lastly, we investigate the benefits of directly learning the eigenvalues in frequency-domain comparing to using a low-rank structure. The model performance in FID along with the computation complexity are shown in in Appendix H, Table 8. As expected, we find that our K-DCT model consistently outperforms "diag+low-rank" models, which in turn outperform purely diagonal models. With increasing rank (until $r = 50$ ($4.8\% of max.$) for CIFAR10 and $r = 100$ ($2.4\%$) for CelebA), the FID decreases but there is still a large gap between low-rank models and our full-rank K-DCT. To understand why, we examined the eigenvalue spectra of the posterior covariances at various points in the sampling process as shown in Appendix H, Fig. 9. These eigenvalues exhibit a power-law decay spanning several ($> 3$) orders of magnitude (especially mid-way through the denoising process), indicating that they cannot accurately be rank-truncated. Notice that the memory consumption of low-rank models increases significantly with higher ranks, while our model keeps negligible memory overhead.

## 5 DISCUSSION & LIMITATIONS

In summary, modeling important elements of natural image statistics can be done in an efficient way through our K-DCT parameterization, and improves image DDPMs especially in the regime of few denoising steps. Given how strongly non-diagonal denoising posterior covariances are (Fig. 1C, top), and how much of this non-diagonal structure the K-DCT model appears to capture (Fig. 1C, bottom), it is perhaps surprising that the performance gains are not more striking; in particular, it is difficult to match FID results from distilled models (e.g. Zhou et al., 2024a; on the other hand, these one-step models lack a tractable likelihood). Perhaps a fundamental limitation of the broader 'covariance modeling approach' is that denoising posteriors in the skip-step regime are far from Gaussian, such that Gaussian sampling is not appropriate even with the right covariance. This problem is analogous to that encountered in second-order optimization, whereby modeling the curvature of the loss function leads to larger, more aggressive parameter updates, but these can easily leave the 'trust region' where the underlying quadratic approximation is valid. In the same way that second-order optimization

benefits strongly from adaptive damping (Martens et al., 2010), second-order sampling might benefit from input-dependent adaptation of the step size (implicitly damping the posterior covariance).

Finally, while the generalizability of our method to non-image data remains an open question, it could potentially be applied to other domains where data exhibits approximate translation invariance, such as audio and speech (see Fig. 18 for a proof of principle on speech data). In fact, perhaps paradoxically, one of the datasets where our K-DCT model performed best (relative to diagonal models) is CelebA, i.e. images of faces that clearly lack translation invariance. Thus, we speculate that K-DCT-like 'full' covariance models that model non-diagonal elements even crudely may lead to performance gains even when the underlying data lacks symmetries.

**Reproducibility statement** Details of model architecture, training and sampling pseudo-code, training and inference details are provided in Appendix F and Appendix G. Notice that, most of the settings, such as *pretrained base models*, are kept same as those used by Bao et al. (2022a); Ou et al. (2024) for fair comparison. Code will be anonymized and uploaded as supplementary materials, and model checkpoints will be realized upon acceptance. We also provide necessary theories and proofs in Appendix E. For compute resources, GPU usage and details of memory/time cost comparisons between different methods are detailed in Table 6. For statistical significance, we provide mean and standard deviation of results using 3 different random seeds in Table 7.

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

# A    USE OF LARGE LANGUAGE MODELS

There is no usage of LLMs throughout this paper.

# B    SKIP-STEP DDPM

Similar to the affine transformation of Eqs. 6 and 7 that relates the predicted mean and covariance of the noise to the 1-step image posterior, we can relate the same quantities to the *skip-step* image posterior $q(\boldsymbol{x}_s|\boldsymbol{x}_t) \approx \mathcal{N}(\boldsymbol{x}_s; \boldsymbol{\mu}_s(\boldsymbol{x}_t), \Sigma_s(\boldsymbol{x}_t))$ for $s < t$, as follows:

$$\boldsymbol{\mu}_s(\boldsymbol{x}_t, t; \theta) = \frac{1}{\sqrt{\bar{\alpha}_{s:t}}} \left( \boldsymbol{x}_t - \frac{1 - \bar{\alpha}_{s:t}}{\sqrt{1 - \bar{\alpha}_t}} \boldsymbol{\epsilon}_\theta(\boldsymbol{x}_t, t) \right) \tag{16}$$

$$\Sigma_s(\boldsymbol{x}_t, t; \phi) = \frac{1}{\bar{\alpha}_{s:t}(1 - \bar{\alpha}_t)} \left( (1 - \bar{\alpha}_{s:t})(\bar{\alpha}_{s:t} - \bar{\alpha}_t)I + (1 - \bar{\alpha}_{s:t})^2 \mathcal{E}_\phi(\boldsymbol{x}_t, t) \right) \tag{17}$$

with the standard notation:

$$\bar{\alpha}_t \triangleq \prod_{t'=0}^{t} \alpha_{t'} \qquad \bar{\beta}_t \triangleq 1 - \bar{\alpha}_t \qquad \bar{\alpha}_{s:t} \triangleq \prod_{t'=s}^{t} (1 - \beta_{t'}) \qquad \bar{\alpha}_{s:t} = \frac{\bar{\alpha}_t}{\bar{\alpha}_s} \tag{18}$$

# C    EXPRESSIVENESS OF THE DCT PARAMETERIZATION

Fig. 3 shows the basis functions from which the spatial (achromatic) component of Eq. 12 is assembled parametrically.

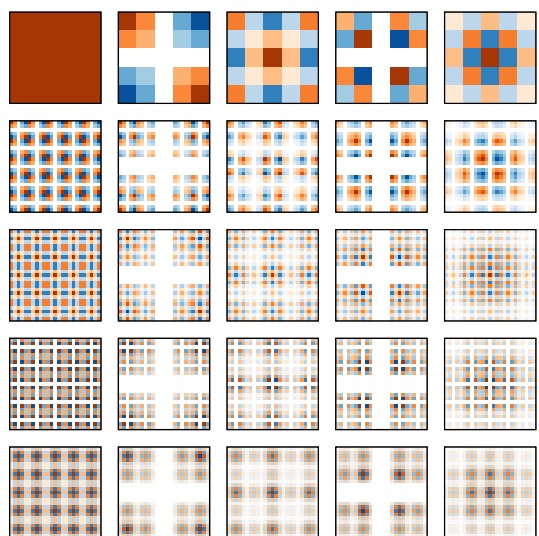

Figure 3: **Expressiveness of the DCT covariance parameterization**. Each panel $(i, j)$ shows $(F \otimes F)^\top \text{diag}(e_{i+dj})(F \otimes F)$ where $F$ is the $d$-points DCT matrix, and $e_k$ is the $k^{\text{th}}$ row of the identity matrix $I_{d^2}$ (here, $d = 5$). Thus, these are the primitive basis functions from which any $(F \otimes F)^\top \text{diag}(\boldsymbol{\lambda})(F \otimes F)$ in Eq. 12 can be assembled.

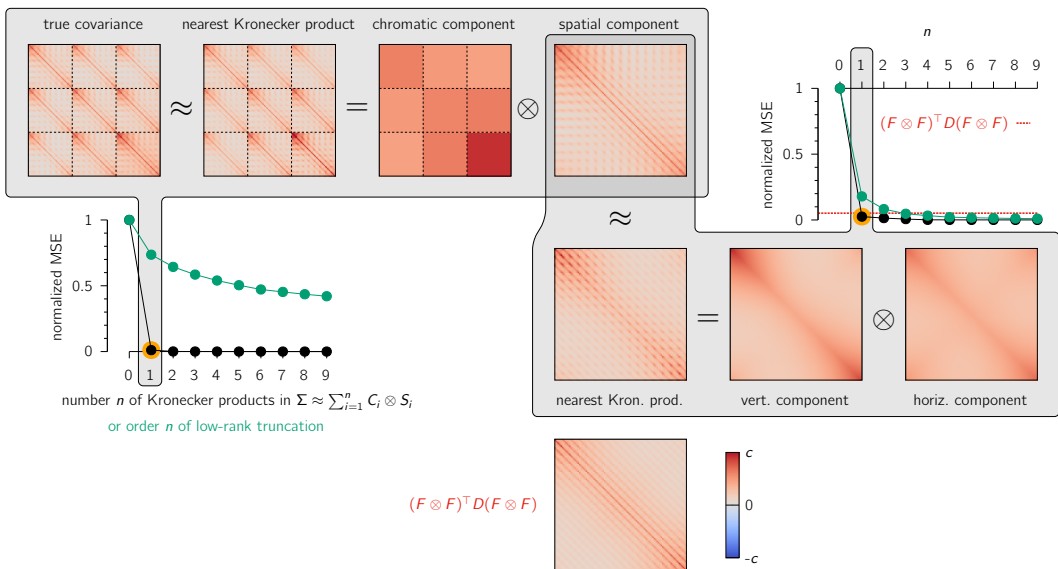

Figure 4: **Accuracy of our covariance parameterization for modeling the marginal (i.e. prior) ImageNet covariance**. **Left**: normalized MSE (black) obtained when approximating the marginal ImageNet covariance $\Sigma$ by a sum of $n$ spatio-chromatic Kronecker products ("colors$(3 \times 3) \otimes$ pixels$(d^2 \times d^2)$"). The optimal decomposition of order $n$ has an analytical form (Van Loan & Pitsianis, 1993). Our approximation in Eq. 12 corresponds to $n = 1$, and is nearly perfect here; this best single Kronecker product approximation is shown at the top, along with the two corresponding factors. We will call the spatial factor $S \in \mathbb{R}^{d^2 \times d^2}$). For comparison, we also show the SVD-based rank-$n$ truncation of $\Sigma$ (green). **Right:** normalized MSE (black) obtained when approximating the spatial factor $S$ (c.f. above) with a sum of vertical-horizontal Kronecker products ($[d \times d] \otimes [d \times d]$). For $n = 1$, this type of approximation is close to, but not exactly the same, as what we propose in Eq. 12 for modeling the spatial component. It would be the same if (i) we constrained our $D \equiv \mathrm{diag}(\boldsymbol{\lambda})$ to itself be a Kronecker product of two smaller diagonals, but (ii) learned the eigenbasis instead of forcing it to be the DCT matrix $F$. Again, a single unconstrained Kronecker product provides an excellent approximation here, indicating that a Kronecker-structured eigenbasis is empirically justified. Moreover, fixing the eigenbasis to $F$ but still learning a full diagonal $D$ (our main proposal) does nearly as well (dashed red; $(F \otimes F)^\top D(F \otimes F)$) whilst having lower computational complexity. **In summary**, this figure shows that **the ImageNet dataset has approximately *separable* spatio-chromatic components**, and is **sufficiently *translation invariant*** for its spatial structure to be compactly described using the DCT.

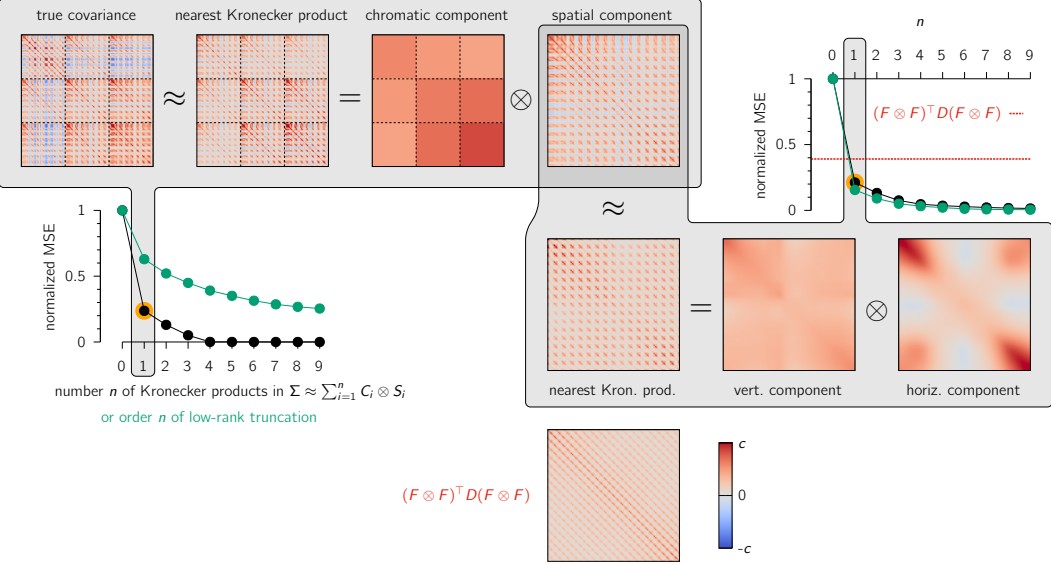

Figure 5: Same as Fig. 4, but for modelling the CelebA marginal (i.e. prior) covariance. Perhaps unsurprisingly, the spatio-chromatic structure of this particular dataset is not as separable as for ImageNet; this is likely due to certain locations in the image being dominated by certain colors (e.g. celebrities aren't known for their blue noses). It is also less translation invariant (i.e. less diagonalizable in the DCT eigenbasis) owing to the nature of these centered portraits.

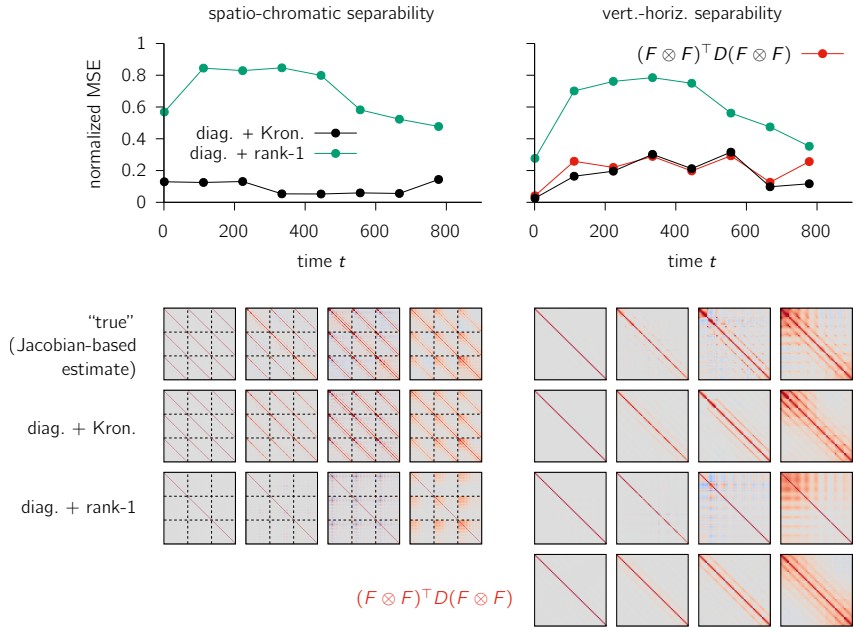

Figure 6: Accuracy of Eq. 12 for modeling conditional covariances at different stages of denoising (time $t$) on CIFAR-10. **Left**: at each time, we have a "true" covariance obtained by differentiating through the score network (c.f. Eq. 2) shown in the first row of images. As in Figs. 4 and 5, for each of these matrices we can find its nearest "diagonal + spatio-chromatic Kronecker product" approximation (second row). This yields a fairly good reconstruction MSE (black), much better than the nearest "diagonal + rank-1" approximation (green; third row of images). Both approximations were obtained through optimization. **Right**: at each time, we can then focus on the spatial component of the Kronecker product identified on the left, and approximate it in the same ways as we did in Fig. 4 (right) with $n = 1$. This is shown with the same color code. In summary, we see that at all times, the use of Kronecker products at both levels outperforms low-rank forms, and that reducing the complexity by restricting the spatial component to a DCT-diagonalized form does not hurt much.

# D POSTERIOR SAMPLING FOR INVERSE PROBLEMS

Our covariance models can be used out-of-the-box in conditional posterior sampling problems such as inpainting. An inverse problem assumes access to some measurements $\boldsymbol{y}$ or corrupted observations of the original input $\boldsymbol{x}_0$. Here we focus on linear-Gaussian observations:

$$\boldsymbol{y} = A\boldsymbol{x}_0 + \boldsymbol{\epsilon}, \ \boldsymbol{\epsilon} \sim \mathcal{N}(0, \Sigma_{\boldsymbol{y}} = \sigma_{\boldsymbol{y}}^2 \boldsymbol{I}) \tag{19}$$

where e.g. $A$ might perform a masking operation. To sample from the posterior distribution $p(\boldsymbol{x}_0|\boldsymbol{y})$, a problem-specific score can be obtained via Bayes' rule as:

$$\nabla_{\boldsymbol{x}_t} \log p_t(\boldsymbol{x}_t|\boldsymbol{y}) = \nabla_{\boldsymbol{x}_t} \log p_t(\boldsymbol{x}_t) + \nabla_{\boldsymbol{x}_t} \log p_t(\boldsymbol{y}|\boldsymbol{x}_t), \tag{20}$$

where the first term is given by the unconditional score network, and the second term represents input-specific guidance. As for unconditional sampling, we make a covariance-based Gaussian approximation, $q(\boldsymbol{x}_0|\boldsymbol{x}_t) = \mathcal{N}(\boldsymbol{x}_0; \mathbb{E}[\boldsymbol{x}_0|\boldsymbol{x}_t], \text{Cov}[\boldsymbol{x}_0|\boldsymbol{x}_t])$ (please refer to Eq. 6 and Eq. 7). Given that the noising process and the observation model are both linear-Gaussian, one can estimate the likelihood score $\nabla_{\boldsymbol{x}_t} \log p_t(\boldsymbol{y}|\boldsymbol{x}_t)$ as (Song et al., 2023; Rozet et al., 2024),

$$\nabla_{\boldsymbol{x}_t} \log p_t(\boldsymbol{y}|\boldsymbol{x}_t) = \nabla_{\boldsymbol{x}_t} \mathbb{E}[\boldsymbol{x}_0|\boldsymbol{x}_t]^\top A^\top (\Sigma_{\boldsymbol{y}} + A\text{Cov}[\boldsymbol{x}_0|\boldsymbol{x}_t]A^\top)^{-1}(\boldsymbol{y} - A\mathbb{E}[\boldsymbol{x}_0|\boldsymbol{x}_t]). \tag{21}$$

When taking smaller steps, using a simple heuristics for $\text{Cov}[\boldsymbol{x}_0|\boldsymbol{x}_t]$ can generate samples with good quality. However, larger steps would require accurate modeling of the covariance and our K-DCT model comes to help. Specifically, our model provides estimates for both $\text{Cov}[\boldsymbol{x}_0|\boldsymbol{x}_t]$ and $\nabla_{\boldsymbol{x}_t} \mathbb{E}[\boldsymbol{x}_0|\boldsymbol{x}_t]^\top$ in the guidance term:

$$\underbrace{\nabla_{\boldsymbol{x}_t} \mathbb{E}[\boldsymbol{x}_0|\boldsymbol{x}_t]^\top}_{\substack{\text{[K-DCT]}=\mathcal{E}_\phi(\boldsymbol{x}_t)/\sqrt{\bar{\alpha}_t} \\ \text{[}\Pi\text{GDM]}=\text{through VJP} \\ \text{[Tweedie's]}=\text{through VJP}}} A^\top (\Sigma_{\boldsymbol{y}} + A \underbrace{\text{Cov}[\boldsymbol{x}_0|\boldsymbol{x}_t]}_{\substack{\text{[K-DCT]}=\bar{\beta}_t \mathcal{E}_\phi(\boldsymbol{x}_t)/\bar{\alpha}_t \\ \text{[}\Pi\text{GDM]}=\bar{\beta}_t \boldsymbol{I} \\ \text{[Tweedie's]}=\text{through VJP}}} A^\top)^{-1}(\boldsymbol{y} - A\mathbb{E}[\boldsymbol{x}_0|\boldsymbol{x}_t]). \tag{22}$$

We compare our method with diagonal covariance modeling OCM (Ou et al., 2024), $\Pi$GDM that uses a heuristic covariance (Song et al., 2023), and a direct evaluation of Tweedie's formula (Rozet et al., 2024) using vector-Jacobian products. We consider a challenging denoising + inpainting painting problem, where $A$ masks out 75% of the pixels uniformly and randomly, with additional *i.i.d.* Gaussian noise ($\sigma_{\boldsymbol{y}} = 10^{-3}$) on the CIFAR10 dataset. Quantitative results are shown in Table 3 and qualitative samples are shown in Fig. 7. Our model has significantly better FID and classification accuracy (AC) when using 10 steps sampling, where all models are tested with the same classification model (pretrained ResNet20).

| Method | Step | FID↓ | AC↑ | Step | FID↓ | AC↑ |
|---|---|---|---|---|---|---|
| K-DCT (ours) | | **14.28** | **72.00**% | | 5.12 | 82.42% |
| $\Pi$GDM (Song et al., 2023) | 10 | 33.89 | 47.48% | 100 | **4.01** | **84.08**% |
| OCM (Ou et al., 2024) | | 77.60 | 36.57% | | 25.93 | 70.95% |
| Tweedie's (Rozet et al., 2024) | | - | - | | 75.21 | 55.16% |

Table 3: Inpainting+denoising results. FID and AC are reported on $50k$ samples.

Notice that Rozet et al. (2024) did not evaluate the vector-Jacobian products on the pretrained unconditional denoiser model, but the improved posterior sampling scheme they proposed can be used to test unconditional diffusion models. Due to numerical errors and imperfect pre-training, the Jacobian matrix is not guaranteed to be perfectly symmetric positive definite, hence our reproduction is far from being superior and we only report for 100 sampling steps. In addition, the inverse operation in Eq. 21 is suggested to be approximated by conjugate gradients (CG) since the covariance matrix should be symmetric positive definite. However, we find CG to perform poorly on our model and we currently choose to compute exact matrix inverse which can be time and memory consuming. Proposing a more efficient approximation method for the inverse is left for future work.

# E LEARNING THE COVARIANCE OF THE NOISE

**Lemma 1** (Use Tweedie's formula to derive the covariance of the noise) Given the DDPM noising process, $q(\boldsymbol{x}_t|\boldsymbol{x}_0) = \mathcal{N}(\boldsymbol{x}_t; \sqrt{\bar{\alpha}_t}\boldsymbol{x}_0 + \bar{\beta}_t I)$, we have that the covariance of the effective noise equals

$$\text{Cov}[\boldsymbol{\epsilon}_t|\boldsymbol{x}_t] = I - \sqrt{\bar{\beta}_t}\nabla_{\boldsymbol{x}_t}\boldsymbol{\epsilon}_\theta(\boldsymbol{x}_t, t) \tag{23}$$

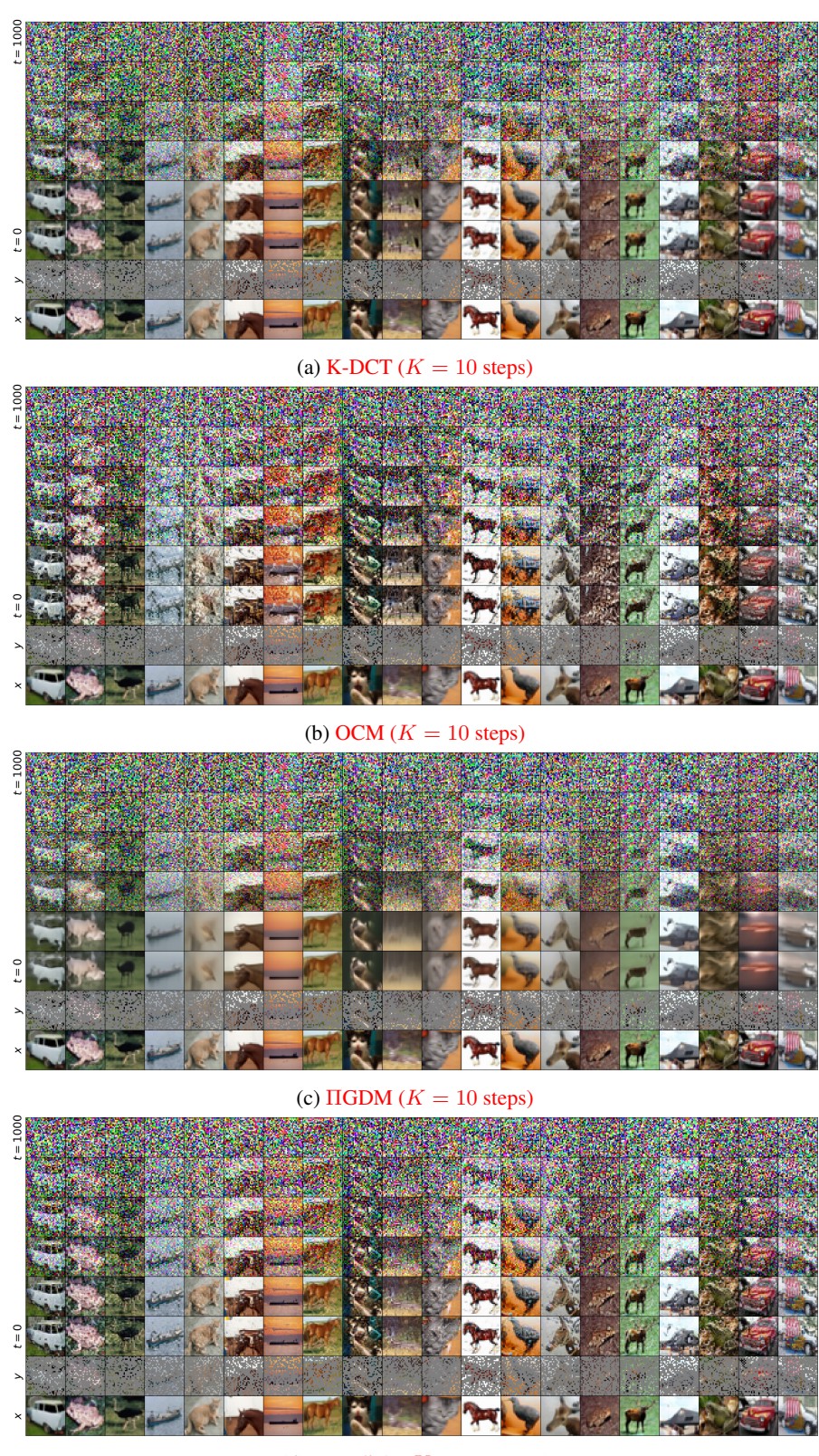

(a) K-DCT ($K = 10$ steps)

(b) OCM ($K = 10$ steps)

(c) ΠGDM ($K = 10$ steps)

(d) Tweedie's ($K = 256$ steps)

Figure 7: Conditional posterior sampling for the noisy inpainting problem, with $\boldsymbol{y} = A\boldsymbol{x} + \sigma_{\boldsymbol{y}}\boldsymbol{\epsilon}$, where $A$ is a random mask that covers 75% of the pixels, and additional Gaussian noise of $\sigma_{\boldsymbol{y}} = 10^{-3}$. Each sub-figure shows conditional denoising sampling procedure for a different method but under the same seed, with the corrupted observations $\boldsymbol{y}$ and the original images $\boldsymbol{x}$ on the last two rows.

*Proof.* Let the perfect noise predictor be $\epsilon_\theta(\boldsymbol{x}_t, t) = \mathbb{E}\left[\frac{\boldsymbol{x}_t - \sqrt{\bar{\alpha}_t}\boldsymbol{x}_0}{\sqrt{\bar{\beta}_t}} | \boldsymbol{x}_t\right]$ which relates to the first order score through $s_1(\boldsymbol{x}_t) \equiv \nabla_{\boldsymbol{x}_t} \log \tilde{q}(\boldsymbol{x}_t) = -\frac{1}{\sqrt{\bar{\beta}_t}}\epsilon_\theta(\boldsymbol{x}_t, t)$. Let the second-order score be $s_2(\boldsymbol{x}_t) \equiv \nabla^2_{\boldsymbol{x}_t} \log \tilde{q}(\boldsymbol{x}_t)$. From Tweedie's first and second order formulae, we have the following:

$$\mathbb{E}[\boldsymbol{x}_0 | \boldsymbol{x}_t] = \frac{\boldsymbol{x}_t + \bar{\beta}_t s_1(\boldsymbol{x}_t)}{\sqrt{\bar{\alpha}_t}} \tag{24}$$

$$\mathrm{Cov}[\boldsymbol{x}_0 | \boldsymbol{x}_t] = \frac{\bar{\beta}_t}{\bar{\alpha}_t}\left(I + (1 - \bar{\alpha}_t)s_2(\boldsymbol{x}_t)\right) \tag{25}$$

From the above equations, one can easily derive,

$$\mathbb{E}\left[\boldsymbol{x}_0\boldsymbol{x}_0^\top - \frac{1}{\sqrt{\bar{\alpha}_t}}(\boldsymbol{x}_0\boldsymbol{x}_t^\top + \boldsymbol{x}_t\boldsymbol{x}_0^\top) \Big| \boldsymbol{x}_t\right] = -\frac{1}{\bar{\alpha}_t}\boldsymbol{x}_t\boldsymbol{x}_t^\top + \frac{\bar{\beta}_t^2}{\bar{\alpha}_t}\left(s_1(\boldsymbol{x}_t)s_1(\boldsymbol{x}_t)^\top + s_2(\boldsymbol{x}_t)\right) + \frac{\bar{\beta}_t}{\bar{\alpha}_t}I \tag{26}$$

Hence, we can solve for the covariance of the added noise through the following derivation:

$$\mathrm{Cov}\left[\frac{\boldsymbol{x}_t - \sqrt{\bar{\alpha}_t}\boldsymbol{x}_0}{\sqrt{\bar{\beta}_t}} | \boldsymbol{x}_t\right]$$

$$= \mathbb{E}\left[\frac{(\boldsymbol{x}_t - \sqrt{\bar{\alpha}_t}\boldsymbol{x}_0)(\boldsymbol{x}_t - \sqrt{\bar{\alpha}_t}\boldsymbol{x}_0)^\top}{\bar{\beta}_t} | \boldsymbol{x}_t\right] - \mathbb{E}\left[\frac{\boldsymbol{x}_t - \sqrt{\bar{\alpha}_t}\boldsymbol{x}_0}{\sqrt{\bar{\beta}_t}} | \boldsymbol{x}_t\right]\mathbb{E}\left[\frac{\boldsymbol{x}_t - \sqrt{\bar{\alpha}_t}\boldsymbol{x}_0}{\sqrt{\bar{\beta}_t}} | \boldsymbol{x}_t\right]^\top$$

$$= \frac{1}{\bar{\beta}_t}\left(\boldsymbol{x}_t\boldsymbol{x}_t^\top + \bar{\alpha}_t\mathbb{E}\left[\boldsymbol{x}_0\boldsymbol{x}_0^\top - \frac{1}{\sqrt{\bar{\alpha}_t}}(\boldsymbol{x}_0\boldsymbol{x}_t^\top + \boldsymbol{x}_t\boldsymbol{x}_0^\top) | \boldsymbol{x}_t\right]\right) - \epsilon_\theta(\boldsymbol{x}_t, t)\epsilon_\theta(\boldsymbol{x}_t, t)^\top$$

$$= \frac{1}{\bar{\beta}_t}\left(\boldsymbol{x}_t\boldsymbol{x}_t^\top + \bar{\alpha}_t\left(-\frac{1}{\bar{\alpha}_t}\boldsymbol{x}_t\boldsymbol{x}_t^\top + \frac{\bar{\beta}_t^2}{\bar{\alpha}_t}\left(s_1(\boldsymbol{x}_t)s_1(\boldsymbol{x}_t)^\top + s_2(\boldsymbol{x}_t)\right) + \frac{\bar{\beta}_t}{\bar{\alpha}_t}I\right)\right) - \epsilon_\theta(\boldsymbol{x}_t, t)\epsilon_\theta(\boldsymbol{x}_t, t)^\top$$

$$= \frac{1}{\bar{\beta}_t}\left(\bar{\beta}_t\epsilon_\theta(\boldsymbol{x}_t, t)\epsilon_\theta(\boldsymbol{x}_t, t)^\top + \bar{\beta}_t^2 s_2(\boldsymbol{x}_t) + \bar{\beta}_t I\right) - \epsilon_\theta(\boldsymbol{x}_t, t)\epsilon_\theta(\boldsymbol{x}_t, t)^\top$$

$$= I + \bar{\beta}_t s_2(\boldsymbol{x}_t)$$

$$= I - \sqrt{\bar{\beta}_t}\nabla_{\boldsymbol{x}_t}\epsilon_\theta(\boldsymbol{x}_t, t) \tag{27}$$

As an alternative to using derivatives of the noise predictor as in Eq. 23 to estimate the covariance of the noise, one can also obtain it as a least-squares estimator in the MMSE approach (c.f. Background). The derivation of this estimator begins with Eq. 26 above, which shows that its r.h.s. is the minimizer of the following mean squared error:

$$\mathbb{E}_{q(\boldsymbol{x}_0)q(\boldsymbol{x}_t | \boldsymbol{x}_0)}\left\|\left(\boldsymbol{x}_0 - \frac{1}{\sqrt{\bar{\alpha}_t}}\boldsymbol{x}_t\right)\left(\boldsymbol{x}_0 - \frac{1}{\sqrt{\bar{\alpha}_t}}\boldsymbol{x}_t\right)^\top - \frac{\bar{\beta}_t^2}{\bar{\alpha}_t}\left(s_1(\boldsymbol{x}_t)s_1(\boldsymbol{x}_t)^\top + s_2(\boldsymbol{x}_t)\right) - \frac{\bar{\beta}_t}{\bar{\alpha}_t}I\right\|_F^2$$

$$= \mathbb{E}_{q(\boldsymbol{x}_0)q(\boldsymbol{x}_t | \boldsymbol{x}_0)}\left\|\frac{\bar{\beta}_t}{\bar{\alpha}_t}\left(\epsilon_t\epsilon_t^\top - I\right) - \frac{\bar{\beta}_t^2}{\bar{\alpha}_t}\left(s_1(\boldsymbol{x}_t)s_1(\boldsymbol{x}_t)^\top + s_2(\boldsymbol{x}_t)\right)\right\|_F^2$$

$$= \mathbb{E}_{q(\boldsymbol{x}_0)q(\boldsymbol{x}_t | \boldsymbol{x}_0)}\left\|\frac{\bar{\beta}_t}{\bar{\alpha}_t}\left(\epsilon_t\epsilon_t^\top - I\right) - \frac{\bar{\beta}_t^2}{\bar{\alpha}_t}\left(\frac{\epsilon_\theta(\boldsymbol{x}_t, t)\epsilon_\theta(\boldsymbol{x}_t, t)^\top}{1 - \bar{\alpha}_t} + s_2(\boldsymbol{x}_t)\right)\right\|_F^2$$

$$= \mathbb{E}_{q(\boldsymbol{x}_0)q(\boldsymbol{x}_t | \boldsymbol{x}_0)}\left\|\frac{\bar{\beta}_t}{\bar{\alpha}_t}\left(\epsilon_t\epsilon_t^\top - \epsilon_\theta(\boldsymbol{x}_t, t)\epsilon_\theta(\boldsymbol{x}_t, t)^\top - \left(I + \bar{\beta}_t s_2(\boldsymbol{x}_t)\right)\right)\right\|_F^2$$

$$= \mathbb{E}_{q(\boldsymbol{x}_0)q(\boldsymbol{x}_t | \boldsymbol{x}_0)}\left\|\frac{\bar{\beta}_t}{\bar{\alpha}_t}\left(\epsilon_t\epsilon_t^\top - \epsilon_\theta(\boldsymbol{x}_t, t)\epsilon_\theta(\boldsymbol{x}_t, t)^\top - \mathrm{Cov}(\epsilon_t | \boldsymbol{x}_t)\right)\right\|_F^2 \tag{28}$$

This derivation shows that the covariance of the noise is the MMSE estimator of $\epsilon_t\epsilon_t^\top - \epsilon_\theta(\boldsymbol{x}_t, t)\epsilon_\theta(\boldsymbol{x}_t, t)^\top$, which leads to the generalized NPR objective below.

# F  EFFICIENT TRAINING & SAMPLING

## F.1  EFFICIENT TRAINING: LOG-LINEAR COMPLEXITY LOSS

When equipped with our parameterization of the posterior covariance in Eq. 12, one can evaluate the loss in Eq. 10 and Eq. 8 in near-linear (in spatial resolution) complexity. Firstly, one can easily extend the objectives for the diagonal case to full covariance. In particular, for NPR, Eq. 10 can be generalized and simplified as follow, (dependency on $(\boldsymbol{x}_t, t)$ is dropped for brevity and the 2D-DCT $F \otimes F$ is also shortened to simply $F$),

$$\mathcal{L}_{\mathrm{NPR}}(\phi) = \mathbb{E}_{q_{\mathrm{data}}(\boldsymbol{x}_0)q(\boldsymbol{x}_t|\boldsymbol{x}_0)}\left[\left\|\mathcal{E}_\phi - (\boldsymbol{\epsilon}_t\boldsymbol{\epsilon}_t^\top - \boldsymbol{\epsilon}_\theta\boldsymbol{\epsilon}_\theta^\top)\right\|_{\mathrm{F}}^2\right] \tag{29}$$

$$= \mathbb{E}_{q_{\mathrm{data}}(\boldsymbol{x}_0)q(\boldsymbol{x}_t|\boldsymbol{x}_0)}\left[\left( \|\mathcal{E}_\phi\|_{\mathrm{F}}^2 + (\boldsymbol{\epsilon}_t^\top\boldsymbol{\epsilon}_t)^2 + (\boldsymbol{\epsilon}_\theta^\top\boldsymbol{\epsilon}_\theta)^2 - 2(\boldsymbol{\epsilon}_t^\top\boldsymbol{\epsilon}_\theta)^2 \right.\right.$$

$$\left.\left. -2\boldsymbol{\epsilon}_t^\top\mathcal{E}_\phi\boldsymbol{\epsilon}_t + 2\boldsymbol{\epsilon}_\theta^\top\mathcal{E}_\phi\boldsymbol{\epsilon}_\theta \right)\right] \tag{30}$$

$$= \mathbb{E}_{q_{\mathrm{data}}(\boldsymbol{x}_0)q(\boldsymbol{x}_t|\boldsymbol{x}_0)}\left[\left( \|\mathrm{diag}(\boldsymbol{\varepsilon}_\phi) + (C_\phi C_\phi^\top \otimes F^\top D_\phi F)\|_{\mathrm{F}}^2 \right.\right.$$

$$-2\boldsymbol{\epsilon}_t^\top\left(\mathrm{diag}(\boldsymbol{\varepsilon}_\phi) + C_\phi C_\phi^\top \otimes F^\top D_\phi F\right)\boldsymbol{\epsilon}_t$$

$$+2\boldsymbol{\epsilon}_\theta^\top\left(\mathrm{diag}(\boldsymbol{\varepsilon}_\phi) + C_\phi C_\phi^\top \otimes F^\top D_\phi F\right)\boldsymbol{\epsilon}_\theta \tag{31}$$

$$\left.\left. + \text{ const. w.r.t. } \phi \right)\right]$$

$$= \mathbb{E}_{q_{\mathrm{data}}(\boldsymbol{x}_0)q(\boldsymbol{x}_t|\boldsymbol{x}_0)}\left[\left( \|\boldsymbol{\varepsilon}_\phi\|_{\mathrm{F}}^2 + \|C_\phi C_\phi^\top\|_{\mathrm{F}}^2 \cdot \|D_\phi\|_{\mathrm{F}}^2 \right.\right.$$

$$+2\mathrm{Tr}\left(\mathrm{diag}(\boldsymbol{\varepsilon}_\phi)\left(C_\phi C_\phi^\top \otimes F^\top D_\phi F\right)\right) \tag{32}$$

$$-2\boldsymbol{\epsilon}_t^\top\left(C_\phi C_\phi^\top \otimes F^\top D_\phi F\right)\boldsymbol{\epsilon}_t - 2\boldsymbol{\varepsilon}_\phi^\top(\boldsymbol{\epsilon}_t \odot \boldsymbol{\epsilon}_t) \tag{33}$$

$$+2\boldsymbol{\epsilon}_\theta^\top\left(C_\phi C_\phi^\top \otimes F^\top D_\phi F\right)\boldsymbol{\epsilon}_\theta + 2\boldsymbol{\varepsilon}_\phi^\top(\boldsymbol{\epsilon}_\theta \odot \boldsymbol{\epsilon}_\theta) \tag{34}$$

$$\left.\left. + \text{ const. w.r.t. } \phi \right)\right]$$

where the trace term (Eq. 32) can be efficiently calculated as follow, (denoting $i$ as pixel in 3D, $c$ as color channel, and $k$ as pixel in 2D),

$$\mathrm{Tr}\left(\mathrm{diag}(\boldsymbol{\varepsilon}_\phi)\left(C_\phi C_\phi^\top \otimes F^\top D_\phi F\right)\right) = \sum_i (\boldsymbol{\varepsilon}_\phi)_i \left(C_\phi C_\phi^\top \otimes F^\top D_\phi F\right)_{ii} \tag{35}$$

$$= \sum_c \sum_k (\boldsymbol{\varepsilon}_\phi)_{ck}(C_\phi C_\phi^\top)_{cc}(F^\top D_\phi F)_{kk} \tag{36}$$

$$= \sum_c (C_\phi C_\phi^\top)_{cc} \sum_k (\boldsymbol{\varepsilon}_\phi)_{ck} \sum_{k'} F_{kk'}^\top (d_\phi)_{k'} F_{k'k} \tag{37}$$

$$= \sum_c (C_\phi C_\phi^\top)_{cc} \sum_k (\boldsymbol{\varepsilon}_\phi)_{ck} \sum_{k'} F_{kk'}^{\top\odot 2} (d_\phi)_{k'} \tag{38}$$

$$= \sum_c (C_\phi C_\phi^\top)_{cc} \left((\boldsymbol{\varepsilon}_\phi)_c^\top F^{\top\odot 2} d_\phi\right) \tag{39}$$

where $[\cdot]^{\odot 2}$ means element-wise square. The above shows that only matrix-vector products and squared Frobenius norm are required for optimizing the objectives. When calculating the matrix-vector product of $\left(C_\phi C_\phi^\top \otimes F^\top D_\phi F\right)\mathrm{vec}(V)$ in Eq. 33 and Eq. 34 for $V = \boldsymbol{\epsilon}_t$ or $V = \boldsymbol{\epsilon}_\theta(\boldsymbol{x}_t, t)$, it can be effectively calculated using FFT as shown in Section 3.2. However, in practice, we find matrix-multiplication to work most efficiently for GPU. The full algorithm of training using the NPR objectives is shown in Algorithm 2.

---

**Algorithm 2** Computing the NPR objective for our parametrization as in Eq. 29

---

**Require:** Covariance model components $\{\varepsilon_\phi, C_\phi, \boldsymbol{d}_\phi\}$, pretrained first-order (noise predictor) model $\epsilon_\theta$, a batch of samples $(\boldsymbol{x}_0, t)$ and a noise scheduler for computing $\alpha_t, \beta_t$.
**Ensure:** $g$ as a batch estimate of Eq. 29
  1: Compute the noised samples, $\boldsymbol{x}_t \leftarrow \sqrt{\bar{\alpha}_t}\boldsymbol{x}_0 + \sqrt{\bar{\beta}_t}\boldsymbol{\epsilon}_t$ with $\boldsymbol{\epsilon}_t \sim \mathcal{N}(0, I)$
  2: Compute model outputs $\boldsymbol{\varepsilon} \leftarrow \varepsilon_\phi(\boldsymbol{x}_t, t)$, $CC^\top \leftarrow C_\phi(\boldsymbol{x}_t, t)C_\phi(\boldsymbol{x}_t, t)^\top$ and $\boldsymbol{d} \leftarrow \boldsymbol{d}_\phi(\boldsymbol{x}_t, t)$
  3: Compute norm $\leftarrow \|\boldsymbol{\varepsilon}\|_\mathrm{F}^2 + \|CC^\top\|_\mathrm{F}^2 \cdot \|\boldsymbol{d}\|_\mathrm{F}^2 + 2\sum_c (CC^\top)_{cc} \langle \boldsymbol{\varepsilon}_c, \text{2D-iDCT}^{\odot 2}(\boldsymbol{d}) \rangle$
      # linear-logarithmic, $\boldsymbol{\varepsilon}_c$ is the $c$-th channel
  4: Define $f(\boldsymbol{v}) = \langle \boldsymbol{v}, \text{2D-iDCT}(\boldsymbol{d} \star \text{2D-DCT}(CC^\top \boldsymbol{v})) + \boldsymbol{\varepsilon} \odot \boldsymbol{v} \rangle$
      # linear-logarithmic, $\star$ is a broadcasting product
  5: Compute trace $\leftarrow f(\boldsymbol{\epsilon}_t) - f(\boldsymbol{\epsilon}_\theta(\boldsymbol{x}_t, t))$
  6: Compute $g \leftarrow$ norm $- 2 \cdot$ trace

---

For OCM, Eq. 8 can be generalized to full covariance (Eq. 23) and approximated using the Hutchinson's trick Hutchinson (1989) as follow, ( $\boldsymbol{v} \sim p(\boldsymbol{v})$ is a Rademacher random variable with entries $\pm 1$, dependency on $(\boldsymbol{x}_t, t)$ is dropped for brevity and the 2D-DCT $F \otimes F$ is shortened to simply $F$),

$$\mathcal{L}_{\text{OCM}}(\phi) = \mathbb{E}_{q_{\text{data}}(\boldsymbol{x}_0)q(\boldsymbol{x}_t|\boldsymbol{x}_0)}\left[\left\|\mathcal{E}_\phi - (I - \sqrt{\bar{\beta}_t}\nabla_{\boldsymbol{x}_t}\boldsymbol{\epsilon}_\theta)\right\|_\mathrm{F}^2\right] \tag{40}$$

$$= \mathbb{E}_{q_{\text{data}}(\boldsymbol{x}_0)q(\boldsymbol{x}_t|\boldsymbol{x}_0)}\left[\|\mathcal{E}_\phi\|_\mathrm{F}^2 - 2\text{Tr}\left(\mathcal{E}_\phi\right) + 2\sqrt{\bar{\beta}_t}\text{Tr}(\mathcal{E}_\phi\nabla_{\boldsymbol{x}_t}\boldsymbol{\epsilon}_\theta) + \text{const. w.r.t. } \phi\right] \tag{41}$$

$$\approx \mathbb{E}_{q_{\text{data}}(\boldsymbol{x}_0)q(\boldsymbol{x}_t|\boldsymbol{x}_0)q(\boldsymbol{v})}\left[\boldsymbol{v}^\top\mathcal{E}_\phi^\top\mathcal{E}_\phi\boldsymbol{v} - 2\boldsymbol{v}^\top\mathcal{E}_\phi\boldsymbol{v} + 2\sqrt{\bar{\beta}_t}\boldsymbol{v}^\top\mathcal{E}_\phi^\top\underbrace{\nabla_{\boldsymbol{x}_t}\boldsymbol{\epsilon}_\theta\boldsymbol{v}}_{\text{JVP}} + \text{const. w.r.t. } \phi\right] \tag{42}$$

where $\mathcal{E}_\phi(\boldsymbol{x}_t, t)\boldsymbol{v}$ can be efficiently evaluated similarly as mentioned when calculating $\mathcal{L}_{\text{NPR}}$, and $\nabla_{\boldsymbol{x}_t}\boldsymbol{\epsilon}_\theta(\boldsymbol{x}_t, t)\boldsymbol{v}$ can be efficiently evaluated using forward-mode AD that does not depend on $\phi$. The full algorithm of training using the OCM objective is shown in Algorithm 3.

---

**Algorithm 3** Computing the OCM objective for our parametrization as in Eq. 40

---

**Require:** Covariance model components $\{\varepsilon_\phi, C_\phi, \boldsymbol{d}_\phi\}$, pretrained first-order (noise predictor) model $\epsilon_\theta$, a batch of samples $(\boldsymbol{x}_0, t)$ and a noise scheduler for computing $\alpha_t, \beta_t$.
**Ensure:** $g$ as a batch estimate of Eq. 40
  1: Sample two random variable, $\boldsymbol{\epsilon}_t \sim \mathcal{N}(0, I)$, $\boldsymbol{v} \sim \text{Bernoulli}(0.5) * 2 - 1$   # Rademacher samples
  2: Compute the noised samples, $\boldsymbol{x}_t \leftarrow \sqrt{\bar{\alpha}_t}\boldsymbol{x}_0 + \sqrt{\bar{\beta}_t}\boldsymbol{\epsilon}_t$
  3: Compute model outputs $\boldsymbol{\varepsilon} \leftarrow \varepsilon_\phi(\boldsymbol{x}_t, t)$, $CC^\top \leftarrow C_\phi(\boldsymbol{x}_t, t)C_\phi(\boldsymbol{x}_t, t)^\top$ and $\boldsymbol{d} \leftarrow \boldsymbol{d}_\phi(\boldsymbol{x}_t, t)$
  4: Compute $\mathcal{E}\boldsymbol{v} \leftarrow \text{2D-iDCT}(\boldsymbol{d} \star \text{2D-DCT}(CC^\top \boldsymbol{v})) + \boldsymbol{\varepsilon} \odot \boldsymbol{v}$
      # linear-logarithmic, $\star$ is a broadcasting product
  5: Compute $H\boldsymbol{v} \leftarrow \text{JVP}(\boldsymbol{\epsilon}_\theta(\cdot, t), \text{primals} = \boldsymbol{x}_t, \text{tangents} = \boldsymbol{v})$   # stop-gradients
  6: Compute $g \leftarrow \langle \mathcal{E}\boldsymbol{v}, \mathcal{E}\boldsymbol{v} \rangle - 2\langle \boldsymbol{v}, \mathcal{E}\boldsymbol{v} \rangle + 2\sqrt{\bar{\beta}_t}\langle \mathcal{E}\boldsymbol{v}, H\boldsymbol{v} \rangle$

---

# G   DETAILS OF EXPERIMENTS

In this section, we provide detailed experimental setup for Section 4, including details for model architectures, training, inference and evaluation. Our setups largely follow those used by (Bao et al., 2022a; Ou et al., 2024) for fair comparison.

**Details of pretrained first-order model**   We have used the same group of pretrained models as in Bao et al. (2022a) and Ou et al. (2024). Table 4 lists the pretrained models and noise schedulers used in our experiments. These models are effectively noise predictors which relates to the first-order score as $\epsilon_\theta(\boldsymbol{x}_t, t) = -\sqrt{\bar{\beta}_t}s_1(\boldsymbol{x}_t)$.

Table 4: Pretrained first-order model used in our experiments, and details of the noise schedulers

| Datasets | Noise scheduler | Sampling steps | Pretrained model |
|---|---|---|---|
| CIFAR10 | Linear | 1000 | Bao et al. (2022a) |
| CIFAR10 | Cosine | 1000 | Bao et al. (2022a) |
| CelebA $64 \times 64$ | Linear | 1000 | Song et al. (2021) |
| ImageNet $64 \times 64$ | Cosine | 4000 | Nichol & Dhariwal (2021b) |

**Details of K-DCT second-order model**   For fair comparison, we follow most of the parameterization as per (Bao et al., 2022a; Ou et al., 2024) for all models. The architecture details of $NN_1$ and $NN_2$ (including three components $\{\varepsilon_\phi, C_\phi, d_\phi\}$) in Eq. 15 are provided in Table 5, where Conv denotes the convolutional layer, Res denotes the residual block for dependence on time $t$, and MLP denotes multi-layer perceptron layers with 1-2 hidden layers. $[\cdot]_{\mathrm{mid}}$ and $[\cdot]_{\mathrm{last}}$ denote positions of the heads, whether they receive output of the UNet from the *middle*-block layer or the *last* up-block layer.

Table 5: Architecture details of parametric heads for the first and second-order model as in Eq. 15

| Datasets | $NN_1$ | $NN_2, \varepsilon_\phi$ | $NN_2, C_\phi$ | $NN_2, d_\phi$ |
|---|---|---|---|---|
| CIFAR10 (LS) | $\mathrm{Conv}_{\mathrm{last}}$ | $\mathrm{Conv}_{\mathrm{last}}$ | $\mathrm{MLP}_{\mathrm{mid}}$ | $\mathrm{MLP}_{\mathrm{mid}}$ |
| CIFAR10 (CS) | $\mathrm{Conv}_{\mathrm{last}}$ | $\mathrm{Conv}_{\mathrm{last}}$ | $\mathrm{MLP}_{\mathrm{mid}}$ | $\mathrm{MLP}_{\mathrm{mid}}$ |
| CelebA $64 \times 64$ | $\mathrm{Conv}_{\mathrm{last}}$ | $\mathrm{Conv}_{\mathrm{last}}$ | $(\mathrm{Res + MLP})_{\mathrm{mid}}$ | $(\mathrm{Res + MLP})_{\mathrm{mid}}$ |
| ImageNet $64 \times 64$ | $\mathrm{Conv}_{\mathrm{last}}$ | $(\mathrm{Res + Conv})_{\mathrm{last}}$ | $(\mathrm{Res + MLP})_{\mathrm{mid}}$ | $(\mathrm{Res + MLP})_{\mathrm{mid}}$ |

**Cost of training and inference time**   In Table 6, we provide empirical comparisons of cost of time for model function evaluation, between the diagonal second-order model and our parameterization of the full covariance model. For training, we provide the average time of one iteration of training update for batch size of 128, which includes evaluation of the corresponding objective, followed by backpropagation. For sampling, we provide the average time of one step of denoising, i.e. calculating $\boldsymbol{\mu}_s(\boldsymbol{x}_t) + \Sigma_s^{1/2}(\boldsymbol{x}_t)\boldsymbol{\xi}$ for $\boldsymbol{x}_t$ of batch size 128. It is clear that our K-DCT model has a negligible additional time compared to diagonal covariances both for training and sampling. As for model memory, the additional memory cost of MLPs for the two additional components, $\{C_\phi, d_\phi\}$, is much smaller than the original UNet.

## G.1   Details of training, inference and evaluation

**Training details**   We use the AdamW optimizer with a learning rate of $1 \times 10^{-4}$ and train for $500K$ iterations across all datasets. The batch size is 128 for all datasets and we select the checkpoint saved every $10K$ iterations with the best FID on $1K$ generated samples with full sampling steps. We train our models using one A6000-48GB GPU for CIFAR10, CelebA, ImageNet; and evaluate on the same machine but one B200-180GB GPU for ImageNet.

Table 6: Averaged time (in millisecond) of one iteration of update w.r.t different objectives (training) and one step of denoising (sampling) for a batch size of 128 on an A6000-48GB GPU

| | Training | | | | Sampling | |
|---|---|---|---|---|---|---|
| | NPR | | OCM | | | |
| Datasets | diagonal | K-DCT | diagonal | K-DCT | diagonal | K-DCT |
| CIFAR10 | 118.74 | 124.82 | 310.02 | 318.89 | 115.74 | 112.49 |
| CelebA $64 \times 64$ | 285.25 | 297.37 | 680.49 | 727.64 | 219.71 | 219.47 |
| ImageNet $64 \times 64$ | 230.34 | 238.16 | 581.51 | 656.89 | 332.61 | 335.63 |

**Sampling details** As all previous papers have noticed, covariance clipping is as crucial to performance as mean clipping in diffusion models. Covariance clipping is trivial in the diagonal case, because at the penultimate sampling step, the condition $\|\Sigma_1(\boldsymbol{x}_2)\|_\infty \mathbb{E}(\boldsymbol{\epsilon}) \leq \frac{2}{255}y$ can be enforced through element-wise clipping on the diagonal. However, for our K-DCT model this would require forming the full covariance matrix . To circumvent this, we propose two methods. The first is applying scaling on both side of the covariance, $S^{1/2}\Sigma_1(\boldsymbol{x}_2)S^{1/2}$, where

$$S^{cij} = \begin{cases} 1, & \text{if } \operatorname{diag}(\Sigma_1)^{cij} \leq e \\ e/\operatorname{diag}(\Sigma_1)^{cij}, & \text{if } \operatorname{diag}(\Sigma_1)^{cij} > e \end{cases} \tag{43}$$

and $e$ is the corresponding threshold. The above only requires access to the diagonal part of the predicted covariance, which is easy and fast to evaluate using the K-DCT structure. The second way is directly clipping the generated sample, $\tilde{\boldsymbol{\xi}} = \Sigma_1(\boldsymbol{x}_2)\boldsymbol{\xi}$, element-wisely by $(-|e\boldsymbol{\xi}|, |e\boldsymbol{\xi}|)$. We find that the second method gives better results. We use the same value of $y$ as in Bao et al. (2022a).

**Evaluation details** All results are evaluated on the exponential moving average of the trained models with a rate of 0.9999. For computing the evidence lower-bound, we calculate the following terms over the entire test set,

$$-L_{\text{ELBO}}(\boldsymbol{x}) = \mathbb{E}_{\text{noising process}}\bigg[\text{KL}(q(\boldsymbol{x}_T|\boldsymbol{x}_0)\|p(\boldsymbol{x}_T))$$

$$+ \sum_{t\in\{k+1:T:k\}} \text{KL}(q(\boldsymbol{x}_{t-k}|\boldsymbol{x}_t,\boldsymbol{x}_0)\|p_{\theta,\phi}(\boldsymbol{x}_{t-k}|\boldsymbol{x}_t)) - \log p_\theta(\boldsymbol{x}_0|\boldsymbol{x}_1)\bigg]. \tag{44}$$

The last sampling step $p(\boldsymbol{x}_0|\boldsymbol{x}_1)$ is approximated by likelihood of discrete image data, which is the same receipt used by Bao et al. (2022a); Ho et al. (2020). Notice that, to calculate the KL terms in Eq. 44, one needs to calculate the (log)-determinant and inverse of $\mathcal{E}_\phi$ which our parameterization does not provide an efficient way of evaluation. The results shown in Table 2 are calculated by forming explicitly the $3D \times 3D$ full matrix which is time consuming but given this is necessary neither for training nor for sampling, we left this for future research. As for computing the FID score, $50K$ samples are generated, whose distribution is compared with the reference distribution statistics using the full training set for CIFAR10 and ImageNet, and $50K$ training samples for CelebA (published by Bao et al. (2022b)). Results in Table 2 are obtained using the same random seed as in (Bao et al., 2022a; Ou et al., 2024). Additionally, variance across three different random seeds is shown in Table 7.

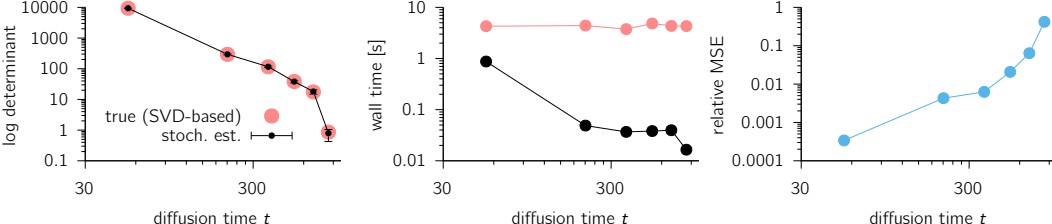

Figure 8: Scalable, efficient evaluation of the log determinant for log-likelihood computations. **Left**: Stochastic estimation (black; mean and quartiles from 100 independent runs) vs. exact value (red) of the log determinant involved in the log-likelihood at different stages of denoising ($t$) on CIFAR10. Here, stochastic estimation is done as shown in Eq. 46 with a *single* probe vector $\boldsymbol{\xi}$. **Center**: Wall time spent in computing the log determinant (CPU). **Right**: Relative MSE in log-determinant estimation, estimated from 100 independent runs. Note that the terms that dominate the log-likelihood (small $t$) are also those that are approximated the best. For large $t$, the noise in Hutchinson's trace estimator with only one probe vector is comparable to the log-det itself, hence the high relative MSE. However, these terms contribute very little to the overall NLL, and they are also the cheapest to compute, so one could use many more probe vectors if accuracy was really required there.

Although for evaluation purposes we computed all log-determinants involved in the KL terms of Eq. 44 by the direct (Cholesky) method, we note that our covariance parameterization affords fast matrix-vector products, and that this property could be used for evaluating log-determinants more efficiently

Table 7: Mean and standard deviation of FID and NLL.

| FID | CIFAR10 (LS) | | | CIFAR10 (CS) | | |
|---|---|---|---|---|---|---|
| # Timesteps $K$ | 10 | 25 | 50 | 10 | 25 | 50 |
| MEAN (K-DCT-NPR) | 22.921 | 9.142 | 5.880 | 17.545 | 7.661 | 5.568 |
| STD (K-DCT-NPR) | 0.154 | 0.019 | 0.040 | 0.032 | 0.108 | 0.011 |
| MEAN (K-DCT-OCM) | 21.980 | 9.082 | 5.914 | 12.865 | 6.298 | 5.092 |
| STD (K-DCT-OCM) | 0.291 | 0.022 | 0.022 | 0.132 | 0.026 | 0.042 |
| | CelebA $64 \times 64$ | | | ImageNet $64 \times 64$ | | |
| # Timesteps $K$ | 10 | 25 | 50 | 25 | 50 | 100 |
| MEAN (K-DCT-NPR) | 19.773 | 13.727 | 9.842 | 23.914 | 18.796 | 17.315 |
| STD (K-DCT-NPR) | 0.090 | 0.080 | 0.038 | 0.093 | 0.131 | 0.135 |
| MEAN (K-DCT-OCM) | 17.457 | 10.873 | 7.447 | 23.812 | 18.833 | 17.262 |
| STD (K-DCT-OCM) | 0.063 | 0.004 | 0.137 | 0.109 | 0.132 | 0.128 |

| NLL | CIFAR10 (LS) | | | CIFAR10 (CS) | | |
|---|---|---|---|---|---|---|
| # Timesteps $K$ | 10 | 25 | 50 | 10 | 25 | 50 |
| MEAN (K-DCT-NPR) | 4.4983 | 4.1076 | 3.9051 | 4.3017 | 3.9853 | 3.8575 |
| STD (K-DCT-NPR) | 0.0028 | 0.0101 | 0.0043 | 0.0032 | 0.0027 | 0.0093 |
| MEAN (K-DCT-OCM) | 4.6079 | 4.1757 | 4.0222 | 4.8201 | 4.2421 | 3.9463 |
| STD (K-DCT-OCM) | 0.0030 | 0.0045 | 0.0026 | 0.0052 | 0.0053 | 0.0054 |
| | CelebA $64 \times 64$ | | | ImageNet $64 \times 64$ | | |
| # Timesteps $K$ | 10 | 25 | 50 | 25 | 50 | 100 |
| MEAN (K-DCT-NPR) | 3.4480 | 3.1714 | 2.9928 | 4.2986 | 4.0770 | 3.9044 |
| STD (K-DCT-NPR) | 0.000649 | 0.000520 | 0.000147 | 0.000406 | 0.000078 | 0.000367 |
| MEAN (K-DCT-OCM) | 3.4383 | 3.1663 | 2.9907 | 4.4180 | 4.1446 | 3.9318 |
| STD (K-DCT-OCM) | 0.000137 | 0.000073 | 0.000086 | 0.000093 | 0.000015 | 0.000051 |

in high-dimension (higher-resolution images). We provide a proof of principle in Fig. 8, using a stochastic estimator the log-determinant based on adaptive numerical quadrature and Hutchinson's trace estimator (Rutten et al., 2020). This estimator is based on the following identity:

$$\log |\Sigma| = \text{Tr}\left[\log \Sigma\right] = \left\langle \boldsymbol{\xi}^{\top} (\log \Sigma) \boldsymbol{\xi} \right\rangle_{\boldsymbol{\xi}} \tag{45}$$

where the expectation is over any spherical distribution $p(\boldsymbol{\xi})$. To compute $(\log \Sigma)\boldsymbol{\xi}$ products, we rely on the integral representation of the matrix logarithm:

$$(\log \Sigma)\boldsymbol{\xi} = \int_0^1 ds (\Sigma - I) \left[s\Sigma + (1-s)I\right]^{-1} \boldsymbol{\xi} \tag{46}$$

For a fixed $\boldsymbol{\xi}$, this can be computed by any numerical quadrature algorithm, using conjugate gradients (CG) to compute the integrand at any $s$ – CG iterations make good use of efficient matrix-vector products. Note that CG at some $s$ can be warm-started by the solution already obtained at another, nearby $s'$. Note also that in early stages of denoising, $\Sigma_t$ can be very ill-conditioned, requiring finer time steps for accurately approximating the integral in Eq. 46. We therefore use an adaptive, Gauss-Kronrod solver, which spends more time near $s = 1$. Finally, we remark that an important property of $\log \Sigma$ is that its eigenvalues have much less spread than those of $\Sigma$ itself. This means that Hutchinson's trace estimator is very accurate even with a single probe vector $\boldsymbol{\xi}$ (Fig. 8).

## H    MORE RESULTS

We carried out comparisons with low-rank methods, in model performance, computation complexity and eigenvalues spectrum analysis in Fig. 9 and Table 8.

Table 8: Model performance and computation complexity between low-rank methods and K-DCT.

| Covariance model | FID | Memory (MB) | Time (ms) |
|---|---|---|---|
| **CIFAR10 (LS)** | | | |
| NPR-diag | 32.35 | 272.39 | 417.66 |
| LowRank(r=5) | 28.81 | 320.46 | 419.68 |
| LowRank(r=10) | 27.57 | 350.52 | 420.52 |
| LowRank(r=25) | 26.79 | 440.72 | 425.36 |
| LowRank(r=50) | 26.14 | 591.05 | 428.15 |
| NPR-KDCT | 23.06 | 298.59 | 420.67 |
| **CelebA** | | | |
| NPR-diag | 28.37 | 552.05 | 1000.55 |
| LowRank(r=5) | 26.15 | 768.11 | 1017.01 |
| LowRank(r=25) | 24.18 | 1224.38 | 1026.02 |
| LowRank(r=50) | 23.35 | 1848.71 | 1039.35 |
| LowRank(r=100) | 23.16 | 3050.05 | 1061.58 |
| NPR-KDCT | 19.69 | 676.81 | 1146.25 |

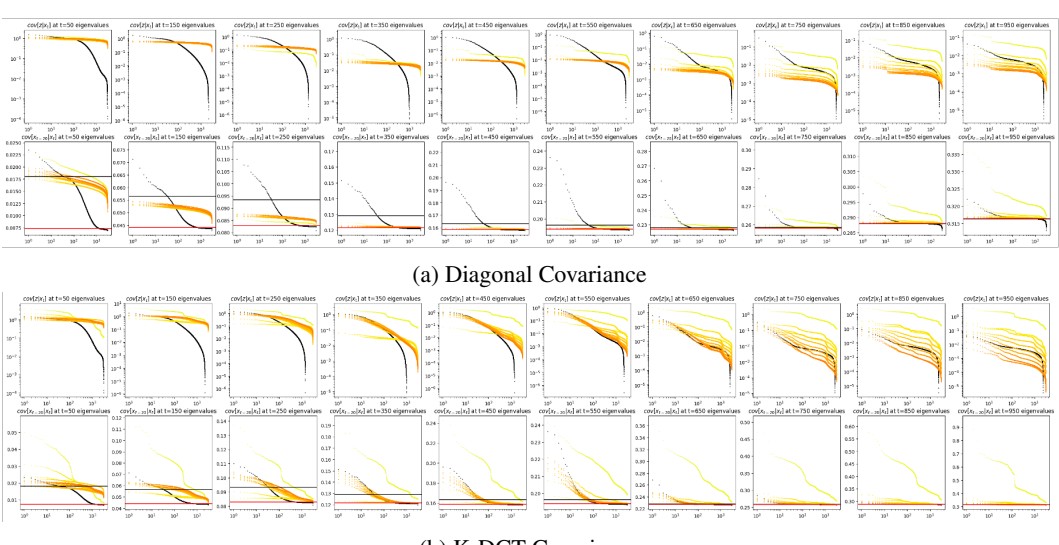

(a) Diagonal Covariance

(b) K-DCT Covariance

Figure 9: Eigen-spectrum of the posterior covariance at various points in the sampling process. Columns from left to right correspond to time from $t = 50$ to $t = T - 50$. Eigenvalues of the **true** posterior covariance are shown in **black** (obtained by evaluating the derivative of the pretrained score network at some random sample as in Eq. 2). Eigenvalues of the fitted covariance with (a) diagonal assumption or (b) our proposed K-DCT structure are shown in orange, where the color from yellow to orange shows the dynamics along training iterations. Within each sub-figure, the first row shows eigenvalues of covariance of the noise, $\mathrm{Cov}[\epsilon_t|\boldsymbol{x}_t]$, while the second row shows eigenvalues of the skip-step covariance used during sampling, $\mathrm{Cov}[\boldsymbol{x}_{t-k}|\boldsymbol{x}_t]$, for $k = 20$. Horizontal lines in red and black indicate the value of 'large', $\beta_t$, and 'small', $\tilde{\beta}_t$, heuristics, respectively.

# I GENERATED SAMPLES

For examples of generated samples with the various methods and noise schedules, see Figs. 10 to 16.

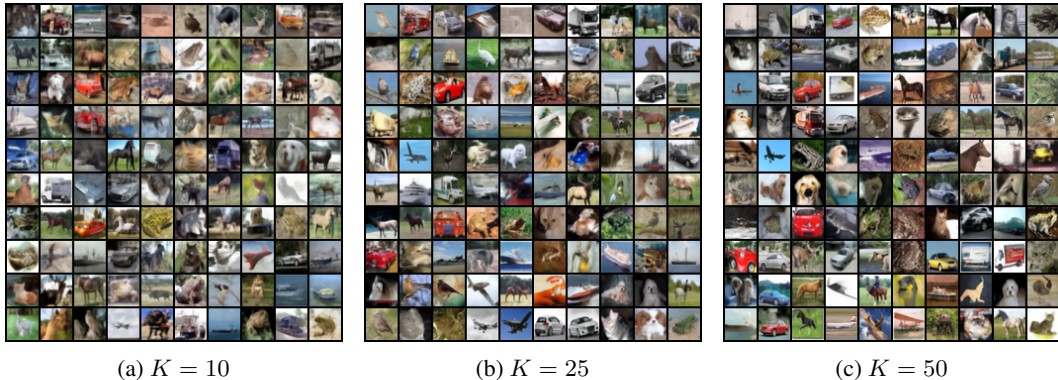

(a) $K = 10$       (b) $K = 25$       (c) $K = 50$

Figure 10: Generated samples with different sampling steps using NPR-K-DCT on CIFAR (LS).

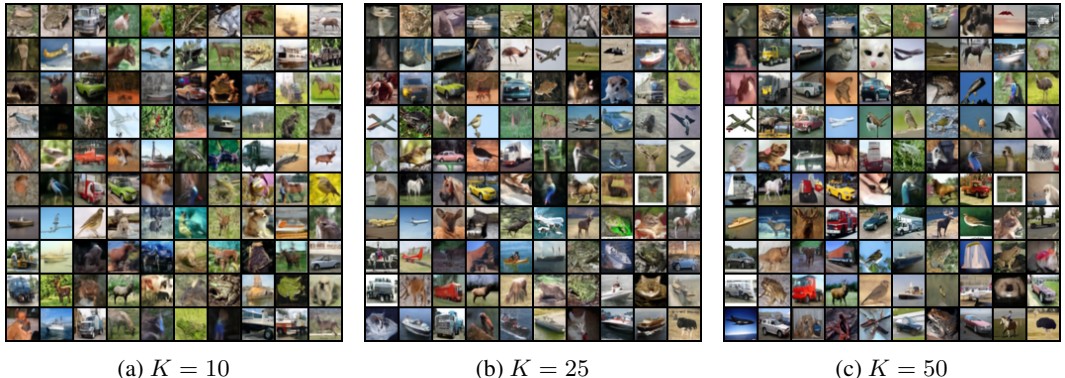

(a) $K = 10$       (b) $K = 25$       (c) $K = 50$

Figure 11: Generated samples with different sampling steps using NPR-K-DCT on CIFAR (CS).

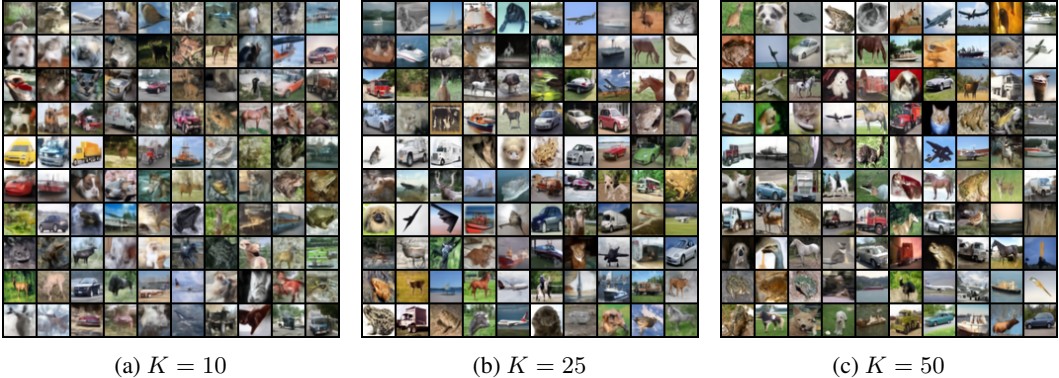

(a) $K = 10$       (b) $K = 25$       (c) $K = 50$

Figure 12: Generated samples with different sampling steps using OCM-K-DCT on CIFAR (LS).

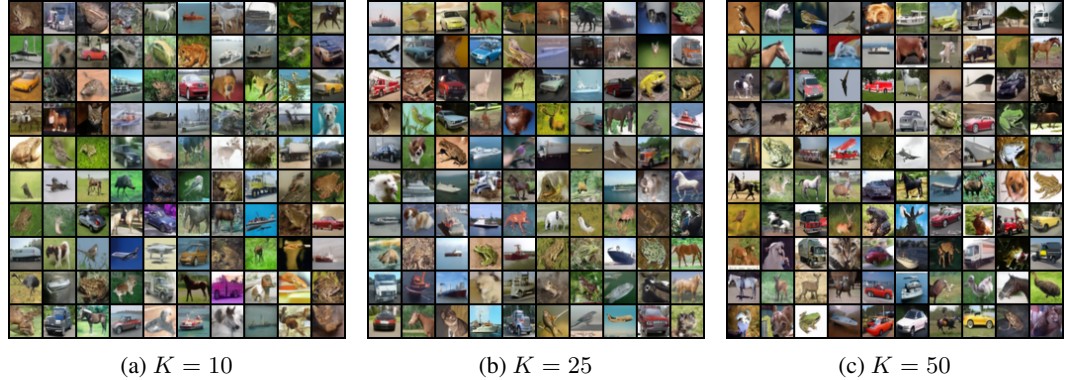

(a) $K = 10$     (b) $K = 25$     (c) $K = 50$

Figure 13: Generated samples with different sampling steps using OCM-K-DCT on CIFAR10 (CS).

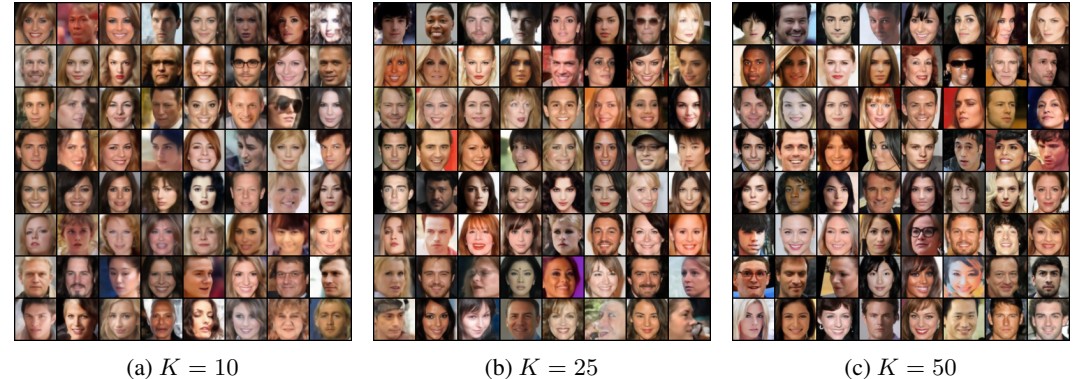

(a) $K = 10$     (b) $K = 25$     (c) $K = 50$

Figure 14: Generated samples with different sampling steps using NPR-K-DCT on CelebA.

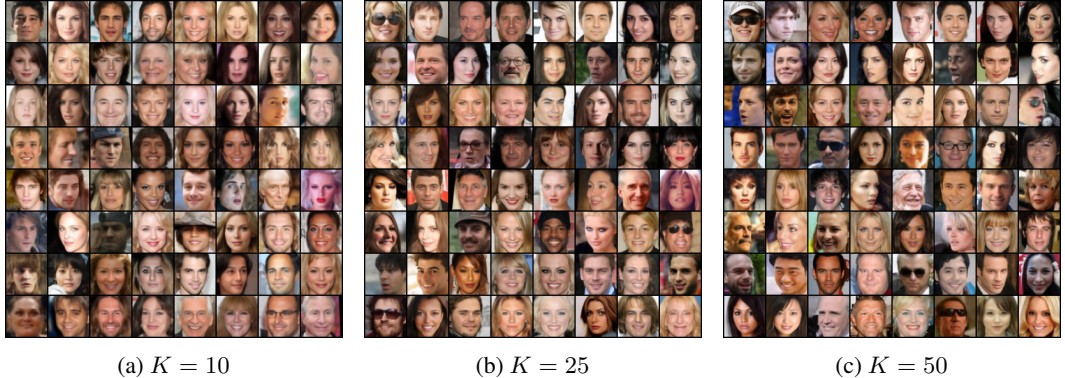

(a) $K = 10$     (b) $K = 25$     (c) $K = 50$

Figure 15: Generated samples with different sampling steps using OCM-K-DCT on CelebA.

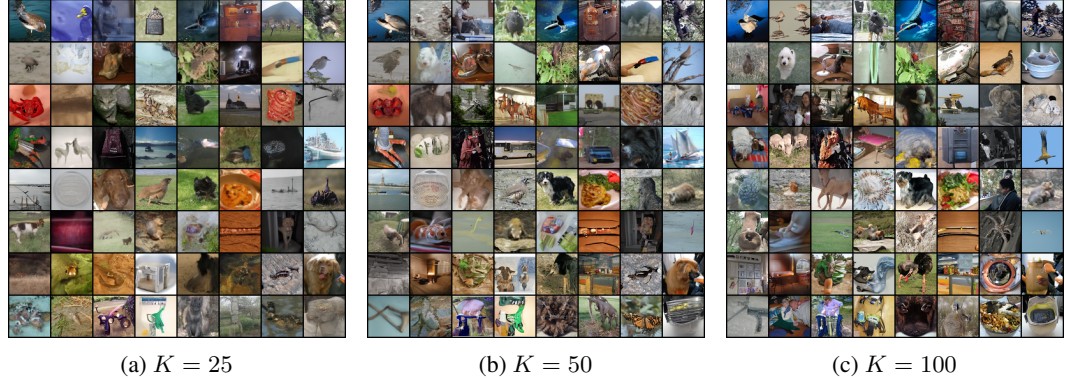

(a) $K = 25$      (b) $K = 50$      (c) $K = 100$

Figure 16: Generated samples with different sampling steps using NPR-K-DCT on ImageNet.

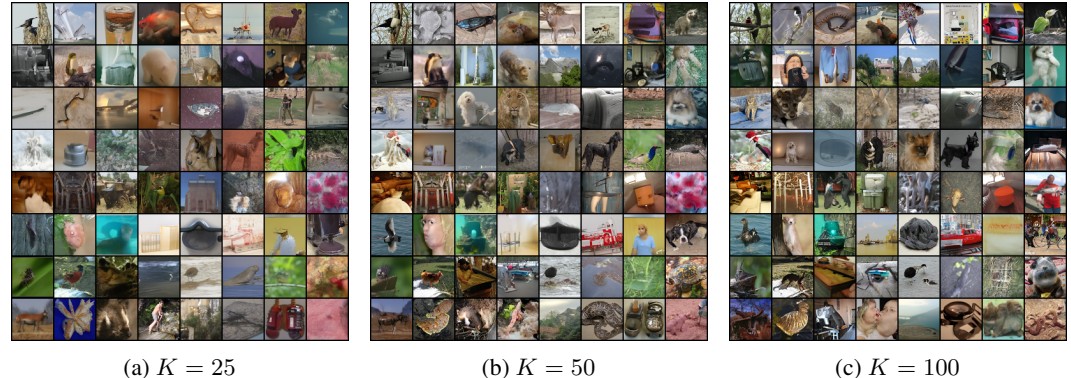

(a) $K = 25$      (b) $K = 50$      (c) $K = 100$

Figure 17: Generated samples with different sampling steps using OCM-K-DCT on ImageNet.

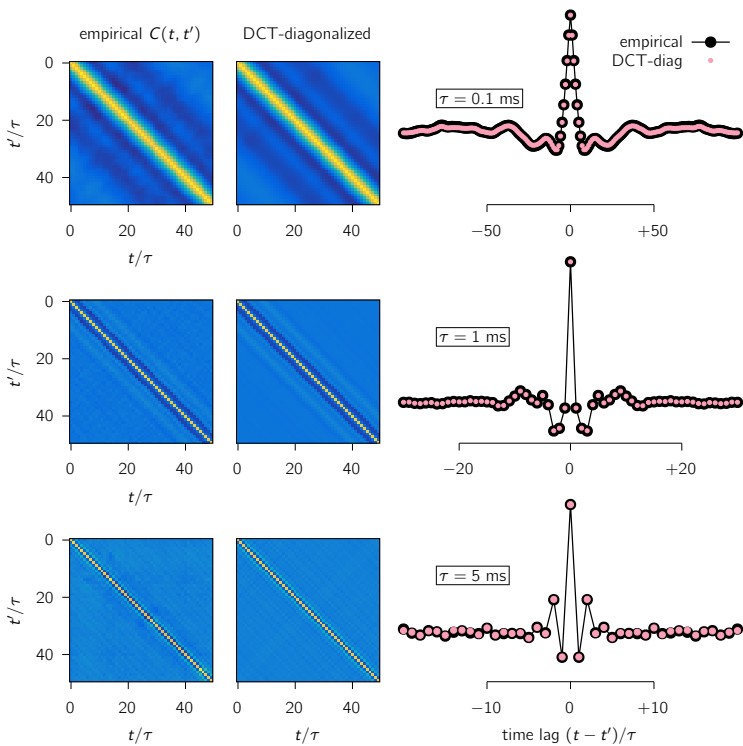

Figure 18: **Accuracy of DCT-based parameterization for audio (speech) data. Left:** Empirical covariance matrix $\Sigma$ of speech (LibriSpeech dataset) at $\tau = 0.1$ ms resolution (top), 1 ms (middle) and 5 ms (bottom). **Center:** Nearest DCT-diagonalized approximation, $\Sigma \approx \hat{\Sigma} \equiv F^\top D F$ where $F$ is the DCT matrix and $D$ is the diagonal matrix that minimizes $\|\Sigma - \hat{\Sigma}\|_F^2$. **Right:** Marginal slices along the secondary diagonal, which collapse those covariance matrices into covariance as a function of time lag $(t - t')/\tau$ assuming stationarity. The DCT parameterization provides a very good fit to the empirical covariance at all resolutions. (Due to limited number of samples in the dataset, covariances on slower timescales ($\tau > 10$ ms) could not be estimated as accurately.)

