# OpenReview forum: "Improved denoising diffusion probabilistic models with efficient non-diagonal covariance modeling"
_ICLR.cc/2026/Conference — Submitted to ICLR 2026_

### Official Review · Reviewer_K15a · 2025-10-19

**Soundness:** 3
**Presentation:** 2
**Contribution:** 3
**Rating:** 4
**Confidence:** 4

**Summary:**

This paper proposes a novel Kronecker-DCT (K-DCT) covariance model for Denoising Diffusion Probabilistic Models (DDPMs) that captures non-diagonal spatio-chromatic correlations in natural images. The model leverages the Discrete Cosine Transform for efficient computation and shows consistent improvements in FID scores and negative log-likelihoods across CIFAR-10, CelebA, and ImageNet datasets compared to diagonal and low-rank baselines, particularly in the few-step sampling regime.

**Strengths:**

[1] The K-DCT formulation creatively combines insights from natural image statistics (separable spatio-chromatic correlations) with efficient computational transforms (DCT), going beyond simple diagonal or low-rank approximations.

[2] The technical execution is thorough, with detailed derivations, efficient implementations, and comprehensive evaluations across multiple datasets and metrics.

[3] The paper is generally well-written with clear explanations of the motivation, method, and results. The appendices provide valuable technical details.

[4] The work addresses the important challenge of accelerating DDPM sampling while maintaining sample quality and likelihood performance, with practical improvements demonstrated on standard benchmarks.

**Weaknesses:**

[1] While consistent, the improvements over diagonal baselines are sometimes incremental (e.g., ~1-2 FID points on CIFAR-10). The paper could better analyze why the significant structural improvements don't translate to larger performance gains.

[2] No comparison against recent distilled models (e.g., MeanFlow, sCM) that achieve superior FID scores, making it difficult to assess the method's position in the current landscape.

[3] The method cannot efficiently compute log-determinants required for exact NLL evaluation (Appendix F.1), limiting its utility for applications requiring precise likelihood calculations.

[4] The discussion of non-Gaussian posteriors in skip-step regimes (Sec. 5) lacks quantitative evidence to support this as a fundamental limitation.

**Questions:**

[1] Given the substantial improvement in covariance structure modeling (Fig. 1C), why are the performance gains over diagonal baselines relatively modest? Are there specific dataset characteristics or sampling regimes where K-DCT provides more substantial benefits?

[2] How does your method compare against recent one-step distilled models in terms of sampling speed vs. sample quality trade-offs? Would incorporating K-DCT covariance modeling benefit distillation approaches?

[3] Have you explored approximate methods for efficient log-determinant computation (e.g., stochastic estimators) that could make exact NLL evaluation feasible without forming the full covariance matrix?

[4] You mention potential application to audio/speech domains - have you conducted any preliminary experiments to validate this? What modifications would be needed for non-image data?

[5] You suggest input-dependent step size adaptation could help with non-Gaussian posteriors - have you experimented with this, and what were the results?

---

> ### Author Response · Authors · 2025-11-24
> **Authors' response - part 1**
>
> Thank you for your thoughtful review. Your comments and questions have led to the addition of 2 new Figures, a new results Table, and several edits in the PDF (detailed below).
>
> 1. “While consistent, the improvements over diagonal baselines are sometimes incremental”.
>
>    This remark was also made by Rev. 6pLT, who prompted us to explore the use of our K-DCT model in more challenging inverse problems (e.g. conditional sampling for in-painting). We have therefore spent some time this week prototyping its use in this setting. We figured that our model can be used out-of-the-box for estimating the “guidance term” in conditional posterior sampling, __with no extra training__. In this context, the K-DCT approach has the main benefits of (i) efficiently and accurately amortizing Tweedie's 2nd-order formula without requiring vector-Jacobian products, and (ii) ensuring positive definiteness (where the Jacobian of the score typically has no such guarantee). In Appendix D of the updated PDF, we describe our strategy for attacking inpainting problems. Specifically, we looked at a challenging inpainting + denoising task, where in addition to iid Gaussian noise corruption, 75% of the pixels are masked out. The results are described in a **new Table 3**, and a **new Figure 7** shows examples of demasked/denoised samples in this task for all four methods.
>
>    The performance improvements of our K-DCT model over ΠGDM (diagonal heuristic, [1]), OCM (diagonal) and direct JVP-based application of Tweedie's formula, are indeed substantial in the few-step regime -- much more pronounced than for unconditional sampling (e.g. FID: 14 (ours) vs. 34 (ΠGDM) for 10 steps; classification accuracy: 72% (ours) vs. 47% (ΠGDM)). We speculate that this may be understood as follows. In unconditional sampling applications, denoising trajectories are predominantly guided by the score, such that covariance information might only be useful in providing greater (and the right type of) sample diversity (reflected in our consistent NLL improvements). In contrast, successful _conditional_ sampling requires appropriately merging likelihood and prior information. When the covariance approximation is too crude (e.g. diagonal), the sampler might either over-trust or under-trust the information which is conditioned upon (i.e.\, the “measurements”). For example, we find that the heuristic covariance model used in ΠGDM [1] tends to under-weigh measurements, resulting in demasked samples that do not align well with the original image (Figure 7b).
>
> 2. “How does your method compare against recent one-step distilled models in terms of sampling speed vs. sample quality trade-offs? Would incorporating K-DCT covariance modeling benefit distillation approaches?”
>
>     Although one-step distillation methods often offer faster generation speed with fewer NFEs compared to sampling-based methods (our focus here), they lack a tractable density and thus present a challenge in applications that require likelihood estimation; these include e.g. importance sampling in AI4Science problems, and diffusion model-based data compression. Our model can provide better likelihood estimates when combined with efficient stochastic estimators (discussed in detail below). Our trained covariance model is a basic building block that could be combined with either distillation approaches or deterministic ODE-solver approaches whenever an accurate model of the conditional covariance is needed. For example, Salimans et al. [2] show that first-order moment-matching already improves distillation. Our model could be used for higher-order moment matching to potentially accelerate distillation training further. As for ODE-solver based methods, Dockhorn et al. [3] goes beyond “second-order in time” and show how “second-order in space” (relying on the Jacobian of the score, i.e. the very object which our K-DCT model amortizes) can improve image generation in the limit of few NFEs. Our model could again act as drop-in replacement for accelerating ODE-solver based model by more accurate modeling of the higher-order information. We appreciate that this is still somewhat speculative at this point, but hope that the reviewer will accept the scope of our paper to be appropriately fenced around sampling-based denoising.
>
> **[ ... continued in next comment ]**

---

> > ### Author Response · Authors · 2025-11-24
> > **Authors' response (ctd.)**
> >
> > 3. “Have you explored approximate methods for efficient log-determinant computation (e.g., stochastic estimators) that could make exact NLL evaluation feasible without forming the full covariance matrix.”
> >
> >     Thank you for raising this point. Yes, stochastic estimators can be used to compute the log-determinant of our covariance model without materialising it. We have added new Figure 8 that demonstrates this. Our K-DCT model affords efficient matrix-vector products which play very well with the stochastic log-det estimator proposed by Rutten et al. [4]. This estimator is based on a combination of Hutchinson's trace estimator ($\log|\Sigma| = \langle \xi^T (\log\Sigma) \xi\rangle_\xi$) and a well-known integral representation of the matrix logarithm ($\log \Sigma = (\Sigma-I) \int_0^1 ds (s\Sigma + (1-s)I)^{-1}$. We detail all this in a new paragraph in Appendix G (new Eqs. 45 and 46). Put together, these result in the following estimator: $\log|\Sigma| \approx \frac1K \sum_{i=1}^K \int_0^1 ds \left[ \xi_i^\top (\Sigma-I) (s\Sigma + (1-s)I)^{-1} \xi_i \right] $. The $[\cdots]^{-1} \xi$ products can be obtained via conjugate gradients, harnessing fast $\Sigma v$ products. We use an adaptive quadrature algorithm to estimate the integral (more work is done near $s=1$). Remarkably, we find that log dets are very well approximated even when using a single probe vector $\xi$ (K=1) in the Hutchinson trace estimator -- likely because the the eigenvalues of $\log(\Sigma)$ have a much smaller spread than those of $\Sigma$ itself -- and the variance of Hutchinson's trace estimator (applied to $\log\Sigma$ here) scales with that spread. We thus confirm that log dets implicated in our CIFAR10 model can be accurately computed orders of magnitude faster than by the direct SVD / Cholesky method (see caption of Figure 8 for additional insights); we of course expect even bigger gains on larger problems. Finally, we note that -- if needed -- this estimator can also be efficiently differentiated using implicit differentiation [4].
> >
> > 4. Potential application to audio/speech domains - have you conducted any preliminary experiments to validate this? What modifications would be needed for non-image data?
> >
> >     This was also raised by Rev. 6pLT. We have taken a look at the autocovariance of speech data (new Figure 18 at the very end of the appendix): no Kronecker factorization is required there (as audio data is inherently 1D, not 3D like images), and a 1D DCT-diagonal model provides a very good covariance approximation over a broad range of timescales. This bodes well for future applications of a DCT-based covariance model for speech diffusion models.
> >
> >
> > 5. “You suggest input-dependent step size adaptation could help with non-Gaussian posteriors - have you experimented with this[...]?”
> >
> >     No, this is still a speculation at this stage, but we do think it will be worth investigating in future work.
> >
> > -----
> >
> > - [1] Song, J., Vahdat, A., Mardani, M., & Kautz, J. (2023, May). Pseudoinverse-guided diffusion models for inverse problems. In International Conference on Learning Representations.
> > - [2] Salimans, T., Mensink, T., Heek, J., & Hoogeboom, E. (2024). Multistep distillation of diffusion models via moment matching. Advances in Neural Information Processing Systems, 37, 36046-36070.
> > - [3] Dockhorn, T., Vahdat, A., & Kreis, K. (2022). Genie: Higher-order denoising diffusion solvers. Advances in Neural Information Processing Systems, 35, 30150-30166.
> > - [4] Rutten, V., Bernacchia, A., Sahani, M. & Hennequin, G. (2020). Non-reversible Gaussian processes for identifying latent dynamical structure in neural data. In NeurIPS.

---

> > > ### Comment · Reviewer_K15a · 2025-11-26
> > >
> > > Thank you very much for your thoughtful and detailed responses to my comments. Your clarifications have addressed several of my concerns, and I appreciate the effort you’ve put into explaining your design choices and experimental setup.
> > >
> > > For future revisions, if possible, it would be extremely helpful if newly added or modified text `could be highlighted`, e.g., in blue or with change tracking, so reviewers can quickly identify what has been updated.
> > >
> > > Below are a few follow-up points based on your rebuttal:
> > >
> > > **Q1**: You attribute the gains to your covariance decomposition strategy. However, similar “no extra training” benefits are commonly achieved via classifier-free guidance (CFG), which is now standard in conditional generation. Could you clarify what specific aspect of your decomposition enables this advantage beyond what CFG already provides?
> > >
> > > Additionally, in Table 3, the dataset name (CIFAR-10) only appears in the caption above the table, not in the table header itself, which caused some initial confusion. More importantly, the reported FID scores appear notably higher than recent SOTA results on CIFAR-10 (even accounting for the inpainting setting). Given that 2025-era diffusion models have pushed `FID well below 2.0 on this dataset`, could you confirm whether your numbers reflect the latest baselines (for example, papers from ICLR, ICML, ICCV, CVPR or NeurIPS2025?) or if there might be a discrepancy in evaluation protocol?
> > >
> > >
> > > **Q2**: Thank you for explaining why one-step methods cannot estimate likelihoods. However, recent ODE-based solvers (e.g., DPM-Solver [1], UniPC [2], and especially EVODiff [3] from this year) achieve high sample quality with NFE < 5 and **do support density estimation via probability flow ODEs**. These methods are now widely adopted thanks to their compatibility with Stable Diffusion and strong speed–quality trade-offs. Yet, they are neither compared against nor mentioned in the paper. This omission makes it difficult to assess the practical relevance of further refining DDPM-style sampling, especially when modern alternatives appear more efficient.
> > >
> > > Moreover, since score-based SDE frameworks (e.g., Song et al.) model noise with isotropic covariance (i.e., scalar noise schedules), the entire trajectory optimization reduces to tuning a 1D schedule, which is far simpler than optimizing a full covariance matrix (which scales quadratically with data dimension). Could you elaborate on why modeling a structured K-DCT covariance is necessary or advantageous in this context?
> > >
> > > **Q3**: I’m glad you tested Hutchinson’s estimator and observed high variance. This aligns with our experience. However, improved variants like Hutchinson++ (with publicly available codes) often yield much more stable and accurate trace estimates at low cost. We’ve found them effective in similar settings. That said, your matrix decomposition approach is indeed interesting and valid; this point is now clarified for me.
> > >
> > > **Q4**: Your explanation here is clear and sufficient. Thank you!
> > >
> > > Overall, while some concerns remain, particularly regarding comparisons with modern fast samplers and the necessity of full covariance modeling. I acknowledge the novelty in your decomposition perspective, but I still tend to keep my score.
> > >
> > > [1] Lu, Cheng, et al. Dpm-solver: A fast ode solver for diffusion probabilistic model sampling in around 10 steps. NeurIPS 2022.
> > >
> > > [2] Zhao, Wenliang, et al. Unipc: A unified predictor-corrector framework for fast sampling of diffusion models. NeurIPS 2023.
> > >
> > > [3] Li, Shigui, et al. Evodiff: Entropy-aware variance optimized diffusion inference. NeurIPS 2025.
> > >
> > > [4] Song, Yang, et al. Score-Based Generative Modeling through Stochastic Differential Equations. ICLR 2021.

---

> > > > ### Author Response · Authors · 2025-11-27
> > > > **thank you for engaging - here are our further responses**
> > > >
> > > > Sorry for not having kept track of the changes in pdf -- the ICLR authors' guidelines suggested this might be done automatically, but apparently not! We have now updated the PDF with necessary highlighted text.
> > > >
> > > > In response to your responses:
> > > > Q1:
> > > > - The mention of 'no extra training required' in our response meant to emphasize that our trained covariance model can be used out-of-the-box for estimating the guidance term in conditional sampling (c.f. Eq.22).
> > > > In other words, a covariance model such as ours is a fairly fundamental building block in diffusion-based applications; here, that includes inverse modelling.
> > > > - Classifier-free guidance is not completely training-free. There is still some, albeit light, training involved; a single neural network parameterizes both the unconditional and the conditional model (the later receiving the class identifier, $y$), and the resulting conditional score must approximate the gradient of $\log p(y|x_t)$ in a Bayesian sense. This means CFG requires training on pairs of $(x_t, y)$ (noisy data and measurements) -- as indeed is discussed in the PiGDM paper (See Table 1 in that paper).
> > > > - All the compared methods in Appendix D are based on PiGDM, which means they are all problem-agnostic -- i.e. they do not require re-training on $(x_t, y)$ pairs). Yet, they still reach the empirical performance of problem-specific ones. Eq.22 in our appendix shows different methods with increasingly reasonable assumptions (and hence better modelling) about the structure of the conditional covariance matrix of $x_0|x_t$.
> > > > - CFG specifically targets class conditions, which is a nonlinear measurement, while our inpainting problems are linear inverse problems with known measurement models (as shown in Eq. 19). Combining our covariance approximation with nonlinear measurement models to tackle nonlinear inverse problem is certainly and interesting future directions, but we do hope the reviewer can understand this is way beyond the scope of our paper.
> > > > - We would genuinely appreciate a reference to any paper that achieves < 2.0 FID on the CIFAR-10 75% masked + noise inpainting task with DDIM style sampling in 10 steps.
> > > > - We strictly followed the code provided in Algorithm 1 of the PiGDM paper, and the results at 100 steps are indeed strong. However, PiGDM performs badly with 10 steps, and the authors themselves acknowledge that their sampler is slow due to vector–Jacobian products through the score network. Our method reduces this overhead and achieves significantly higher classification accuracy under the same 10-step budget.
> > > >
> > > > Q2:
> > > > - The recent ODE-based solvers you mentioned, such as DPM-solvers and EVODiff, do not themselves come with density estimation. A more related paper might be [1], where several techniques are introduced to improve likelihood estimates for diffusion ODEs. These include velocity parameterization and importance sampling for variance reduction, but still require Hutchinson's trace estimator and Jacobian-vector products. We acknowledge this line of work, but we still stand by our method for providing more straightforward likelihood estimation.  With a proper stochastic estimator, we obtain low-variance (c.f. below) estimates at very reasonable compute and memory cost, without requiring further engineering work.
> > > > - The whole premise of our paper is that, when taking larger sampling steps, the isotropic covariance assumption is no longer valid -- such that we cannot reduce the denoising problem to a mere optimization of a noise schedule! This isotropic assumption is precisely the one made by methods labelled ``Heuristic large/small'' in all of our tables. DPMs can be easily extended to continuous SDEs, as fully discussed in detail in section 5 of ref [2], but it suffers a lot when taking larger stepsize. It is exactly when a better model of the conditional covariance model comes to help. A key point of our paper is that we manage to model the covariance matrix in “full” (non-diagonal) form, but without ever having to materialise it. All of our operations, during __training__, __sampling__ for evaluation and now even likelihood estimation, are based on efficient matrix-vector products that only scale linearly with the data dimension. Our method is never quadratic in memory or complexity, yet allows second-order information to be incorporated in the sampling process.
> > > >
> > > > Q3:
> > > > - To be clear that, on the contrary, Hutchinson's estimator with our adpative numerical quadrature has __low__ variance. As shown in Figure. 8, where only __single__ probe vector is used, we achieve below $10^-3$ relative MSE for the hardest timestep -- and that is the log-det that dominates the NLL by far.
> > > >
> > > >
> > > > [1] Improved Techniques for Maximum Likelihood Estimation for Diffusion ODEs
> > > > [2] Estimating the Optimal Covariance with Imperfect Mean in Diffusion Probabilistic Models

---

### Official Review · Reviewer_ZmJB · 2025-10-30

**Soundness:** 3
**Presentation:** 3
**Contribution:** 3
**Rating:** 6
**Confidence:** 4

**Summary:**

This paper proposes a covariance parameterization for DDPMs, called Kronecker-DCT (K-DCT).The main idea is to better approximate the posterior covariance in DDPMs by explicitly modeling the non-diagonal spatial and chromatic correlations inherent in natural images. The approximately separable chromatic and spatial correlations permit a Kronecker factorization, with the spatial component efficiently modeled in the DCT frequency domain under approximate translational invariance. This structure reduces the computational complexity of full covariance modeling from O(D^2) to O(D \log D), allowing efficient training and sampling. Experiments on CIFAR-10, CelebA, and ImageNet 64×64 datasets show consistent improvements in both FID and negative log-likelihood (NLL) metrics compared to diagonal and low-rank covariance baselines.

**Strengths:**

1.	The K-DCT parameterization is a good way to model non-diagonal correlations. The combination of Kronecker factorization and DCT-based representation is efficient.
2.	Despite its more complex formulation, the proposed model retains log-linear training and sampling complexity by exploiting the Kronecker and DCT factorizations , having efficient computation of matrix–vector products and DCT-based transformations.
3.	This study evaluates the proposed method on multiple datasets with both linear and cosine noise schedules, comparing six covariance modeling variants. Results consistently show superior FID and NLL performance, especially in the skip-step regime.

**Weaknesses:**

1.	The K-DCT covariance model fundamentally relies on the assumption that chromatic and spatial correlations are approximately separable and translation-invariant. This assumption justifies the Kronecker-DCT term but is only qualitatively motivated by natural image statistics and a few visual examples. However, the paper does not quantitatively evaluate the validity of this assumption across datasets or denoising stages. The generalizability of this structured covariance form remains uncertain.
2.	Experiments are restricted to low resolution image datasets and the approach has not been tested on higher-resolution (≥256²) settings or within latent-diffusion frameworks. Moreover, generalization beyond the image domain (e.g., audio) is left unverified, which the authors themselves acknowledge as an open question.
3.	Eq.12 introduces three distinct components—diagonal compensation, color-channel correlation, and frequency-domain spectrum—but the paper provides no ablation to isolate their contributions. As a result, it remains unclear which component contributes most to FID/NLL improvements, and whether simpler variants would yield comparable gains.

**Questions:**

1.	Have the authors quantitatively evaluated how well the separability and approximate translation-invariance assumptions hold across different datasets?
2.	Have the authors evaluated whether the approximate translation-invariance assumption remains valid across different denoising timesteps? Since the spatial correlation structure changes as noise decreases, is this assumption still reasonable in both early and late diffusion stages?
3.	Can the proposed K-DCT covariance model be efficiently extended to higher-resolution datasets or latent-diffusion frameworks such as Stable Diffusion?
4.	How do these components interact during training—do they capture complementary statistical structures, or is there redundancy among them?

---

> ### Author Response · Authors · 2025-11-24
> **Authors' response**
>
> We are grateful for your time and effort reviewing our submission. Your comments and questions have led to the addition of 3 new Figures, and several edits in the PDF (detailed below).
>
> 1. You wanted to see a more quantitative assessment of spatio-chromatic separability and DCT-diagonalisability, across different dataset and across various denoising timesteps. This was also raised by Rev. 6pLT. We have therefore **added three new Figures** (4, 5, and 6 in the revised PDF's appendix) where we carefully tease apart how the statistical structure of images warrants each key aspect of Eq. 12. These results show that spatio-chromatic separability is very strong in ImageNet, and somewhat less strong in CelebA (for very intuitive reasons; e.g. a celebrity's nose is always found in the center of the image, but is rarely blue). In both cases though, the approximation obtained by Kronecker factorization of color and pixel spaces is vastly more economical than using standard low-rank truncation. We also show that these image datasets are sufficiently translation invariant for the spatial (pixel space) component of their covariance to be well approximated in diagonal form in the DCT eigenbasis (again better than low-rank truncation). These results hold not only for marginal (prior) covariances in large datasets (Figures 4-5), but also for the conditional covariances that arise at various stages of denoising on CIFAR10 (Figure 6).
>
> 2. You wondered if our K-DCT covariance model might be efficiently extended to higher-resolution datasets, or to latent-diffusion frameworks such as Stable Diffusion.
>    A key benefit of the K-DCT model is its parameter efficiency, and the fact that although the covariance is modelled in full (as opposed to diagonal form), it needs not be materialized, resulting in negligible memory and compute overhead in both training and inference (point 3. in our response to Rev. K15a now also shows that log-likelihood calculations can be performed efficiently, too!). In this respect, our model is just as efficient as models that use a diagonal model for the covariance. Whilst, technically, the K-DCT parameterization can be applied to latent-space DPMs, we cannot think of any a priori justification for using this structure to model the posterior covariance in latent space. In fact, for latent models where the map from pixels to latent space already performs an approximate frequency decomposition, the posterior covariance may not benefit as much from more structured modelling beyond diagonal.
>
> 3. You wondered how the various components of our covariance model interact during training; and in particular, you suggested performing ablations to assess whether they capture complementary / redundant statistical structures.
>
>     Before landing on our final proposed model in the form of Eq.12, we had tested several alternatives, such as one without diagonal compensation (easier to invert for e.g. NLL calculations), and one with a 3-way Kronecker product (color channels $\otimes$ vertical $\otimes$ horizontal) where the last two factors were not constrained to the DCT eigenbasis. We found diagonal compensation to be essential (without it, results are much worse). Moreoever, the 3-way Kronecker form proved significantly harder to train (slower/never converge to similar training loss as our K-DCT formulation), possibly due to increased parameter degeneracies introduced by the multiplication of three different free parameters. We have not had the chance to go back to these early explorations this week, but we will be adding a supplementary section with details of such ablations in the final submission.
>
>     We also hope that our response to point 1. above further clarifies the differential role of the various structures in Eq. 12 in modelling separate aspects of image statistics.

---

### Official Review · Reviewer_CmXv · 2025-10-31

**Soundness:** 2
**Presentation:** 4
**Contribution:** 3
**Rating:** 8
**Confidence:** 3

**Summary:**

This paper introduces a novel approach for parametrizing a non-diagonal covariance matrix in the reverse process of Gaussian diffusion models. The proposed parametrization draws on inductive biases from computer vision related to correlations between pixels and channels in natural images. The authors further develop an efficient training and sampling methodology, building in part on prior advancements aimed at enhancing diffusion models. Experimental results demonstrate the effectiveness of the proposed method within the standard stochastic sampling procedure of classical DDPM. Comparisons are conducted against existing models employing various covariance parametrization strategies.

**Strengths:**

- The paper is clearly written, with well-motivated arguments and logical structure.

- The proposed training and sampling procedures make the theoretical contributions practically applicable.

- The approach has strong potential to become a commonly adopted component in future work.

- Experimental results clearly highlight the impact of the proposed method.

**Weaknesses:**

While the experimental comparison is fair, the standard DDPM samplers under few-step sampling regimes perform significantly worse than ODE-based or consistency-based samplers. Additional experiments demonstrating the effectiveness of the proposed parametrization in such alternative settings would further strengthen the contribution and soundness of the paper.

**Questions:**

See the Weaknesses section.

---

> ### Author Response · Authors · 2025-11-24
> **Authors' response**
>
> Thank you for your very positive appraisal of our work.
>
> > While the experimental comparison is fair, the standard DDPM samplers under few-step sampling regimes perform significantly worse than ODE-based or consistency-based samplers.
>
> We agree that, in the current state of affairs in denoising, the best strategies are those that do not rely on sampling (ODE-based, consistency-based samplers, ...). An important extension of our work will therefore be to deploy our covariance parameterization to further improve these deterministic techniques. Indeed, the K-DCT model is a basic building block that could be combined with either distillation approaches or deterministic ODE-solvers wherever an accurate model of the conditional covariance is needed. For example, Salimans et al. [1] show that first-order moment-matching already improves distillation. Our model could be used for higher-order moment matching to potentially accelerate distillation training further. As for ODE-solver based methods, Dockhorn et al. [2] goes beyond “second-order in time” and show how “second-order in space” (relying on the Jacobian of the score, i.e. the very object which our K-DCT model amortizes) can improve image generation in the limit of few NFEs. Our model could again act as a drop-in replacement for accelerating ODE-solver based model by more accurate modeling the required higher-order information. We appreciate that this is still somewhat speculative at this point, but we are glad that the reviewer accepts the scope of our paper to be appropriately fenced around sampling-based denoising.
>
> - [1] Salimans, T., Mensink, T., Heek, J., & Hoogeboom, E. (2024). Multistep distillation of diffusion models via moment matching. Advances in Neural Information Processing Systems, 37, 36046-36070.
> - [2] Dockhorn, T., Vahdat, A., & Kreis, K. (2022). Genie: Higher-order denoising diffusion solvers. Advances in Neural Information Processing Systems, 35, 30150-30166.

---

### Official Review · Reviewer_6pLT · 2025-10-31

**Soundness:** 3
**Presentation:** 3
**Contribution:** 2
**Rating:** 4
**Confidence:** 4

**Summary:**

This paper introduces a new method of parameterizing and learning denoiser covariance when training diffusion models. While previous work assumed diagonal and/or low rank formulations, this work used a custom covariance model motivated from empirical observations on image datasets.

Overall this method is a bit ad-hoc and unclear if there is a general principle that can be applied to other data modalities, and its application to unconditional sampling does not seem to improve over state-of-the-art.

**Strengths:**

This paper is clear and well-written. It motivates a new covariance parameterization taking into account natural image statistics which improves over previous proposed parameterizations. There are also comprehensive experiments evaluating its performance for unconditional diffusion sampling.

**Weaknesses:**

1. Although this covariance model fits better to natural image statistics, it is still hand-designed and unclear if it will generalize to other data modalities. It would be better if this parameterization is directly learned from data, or if there is a way to directly construct it from measured data statistics.
2. The experimental evaluations mainly focuses on few-step unconditional sampling, but it is unclear how much unconditional sampling benefits from posterior covariance estimation. The numerical FID results are better than the diagonal-covariance baseline, but do not seem to be better than state of the art methods such as EDM [Karras et. al. 2022].

**Questions:**

1. It would strengthen this paper if this particular method of covariance estimation is applied to using diffusion models to solve inverse problems. In this setting adding noise during the denoising process is more important and there could be potentially bigger improvements than in unconditional sampling.
2. How would the generation results in Table 2 compare to DDIM sampling?
3. Line 215: What is the $||\cdot||_F$ term? In Eq. 8, does "diag" extract the diagonal of a matrix?
4. Is the $(\cdot)^2$ in Eq. 10 acting element-wise on the vector?

---

> ### Author Response · Authors · 2025-11-24
> **Authors' response**
>
> Thank you for your time and effort assessing our paper. Your comments and questions have led to the addition of 5 new Figures, a new table of results, and to several edits in the PDF (detailed below).
>
> 1. Regarding your concern that our covariance parameterization is “hand-designed”, and the open question of whether it would generalize to other data modalities:
>     - This is a valid concern, but we want to emphasize that our model expresses well-known aspects of natural images statistics, such as spatio-chromatic separability and translation-invariance. When considering these features, both the Kronecker product structure and the use of a DCT basis in Eq. 12 arise as very natural choices. Thus, whilst this is still “hand-designed”, the design space is very constrained by what we know about natural images.
>
>         Nevertheless, we agree that we could have provided a more detailed assessment of whether the assumptions behind Eq 12 of spatio-chromatic separability and DCT-diagonalisability hold in practice. This is also something ZmJB thought would strengthen the paper. We have therefore **added three new Figures** (4, 5, and 6 in the revised PDF's appendix) where we carefully tease apart how the statistical structure of images warrants each key aspect of Eq. 12. These results show that spatio-chromatic separability is very strong in ImageNet, and somewhat less strong in CelebA (for very intuitive reasons; e.g. a celebrity's nose is always found in the center of the image, but is rarely blue). In both cases though, the approximation obtained by Kronecker factorization of color and pixel spaces is vastly more economical than using standard low-rank truncation. We also show that these image datasets are sufficiently translation invariant for the spatial (pixel space) component of their covariance to be well approximated in diagonal form in the DCT eigenbasis (again better than low-rank truncation). These results hold not only for marginal (prior) covariances in large datasets (Figures 4-5), but also for the conditional covariances that arise at various stages of denoising on CIFAR10 (Figure 6).
>
>     - As for other data modalities (also asked by Rev. K15a), we have taken a look at the autocovariance of speech data (**new Figure 18** at the very end of the appendix): no Kronecker factorization is required there (as audio data is inherently 1D, not 3D like images), and a 1D DCT-diagonal model provides a very good covariance approximation over a broad range of timescales. This bodes well for future applications of a DCT-based covariance model for speech diffusion models.
>
> 3. You made a very useful remark, namely that our covariance estimation strategy might yield greater benefits when applied to _conditional_ sampling problems -- something we hadn't explored. We have therefore prototyped its use in inverse problems, as you suggested. We figured that our model can be used out-of-the-box for estimating the “guidance term” in conditional posterior sampling, __with no extra training__. In this context, the K-DCT approach has the main benefits of (i) efficiently and accurately amortizing Tweedie's 2nd-order formula without requiring vector-Jacobian products, and (ii) ensuring positive definiteness (where the Jacobian of the score typically has no such guarantee). In Appendix D of the updated PDF, we describe our strategy for attacking inpainting problems. Specifically, we looked at a challenging inpainting + denoising task, where in addition to iid Gaussian noise corruption, 75% of the pixels are masked out. The results are described in a **new Table 3** (FID: 14 (ours) vs. 34 (ΠGDM) for 10 steps; classification accuracy: 72% (ours) vs. 47% (ΠGDM)). The performance improvements of our K-DCT model over ΠGDM (diagonal heuristic), OCM (diagonal) and direct AD-based application of Tweedie's formula, are indeed substantial in the few-step regime (much more pronounced than for unconditional sampling, as you correctly intuited!). In a new Figure 7, we provide examples of demasked/denoised samples in this task for all four methods.
>
> 4. Answers to your minor queries:
>
>    - The samples in Appendix D are generated with DDIM sampling as baseline (apart from the guidance term).
>    - The term $\|\cdot\|_F^2$ in Line 215 referred to the Frobenius norm in Eq. 29 in Line 983. We agree this was very confusing; we have therefore rewritten this entire paragraph to improve clarity (including defining diag, which indeed extracts the diagonal of a matrix).
>    - Yes, in Eq.10, $(\cdot)^2$ acts element-wise; we have clarified this.

---

### Comment · Area_Chair_p53h · 2025-11-27

Dear reviewers,

Please review the rebuttal and discuss with the authors if you have not done it.

Thanks,
AC

---

### Meta-Review · Area_Chair_pmLi · 2026-01-06

**Summary:**

The reviews are mixed. One reviewer recommends acceptance (rating 8), but the review is relatively sparse and does not provide enough technical justification to outweigh the remaining concerns. Reviewer ZmJB is moderately positive (rating 6), yet their main points align with the more critical reviews.

The central issue is that the proposed K-DCT covariance parameterization relies on a strong structural and application assumption: that chromatic and spatial correlations are approximately separable and translation-invariant. This makes the model largely hand-designed, and it remains unclear how well it would generalize beyond the specific image regimes studied, or to other data modalities. Several reviewers also noted that it would be more compelling if the parameterization were learned directly from data, or if there were a principled way to construct it from measured data statistics, rather than fixing the form a priori.

Given these limitations, the contribution appears primarily engineering-driven, with incremental empirical improvements and limited conceptual novelty and uncertain applicability. Considering ICLR’s emphasis on broadly impactful and principled advances, it is difficult to find the current version sufficiently compelling.

**Reviewer Concerns:**

.

**Reviewer Scores:**

.

---

### Decision · Program_Chairs · 2026-01-26

Reject